# Labrador Sea sub-surface density as a precursor of multi-decadal variability in the North Atlantic: a multi-model study.

Pablo Ortega[1,2], Jon I. Robson[1], Matthew Menary[3], Rowan T. Sutton[1], Adam Blaker[4], Agathe Germe[4], Jöel J.-M. Hirschi[4], Bablu Sinha[4], Leon Hermanson[5,] and Stephen Yeager[6]

[1] NCAS, University of Reading, Reading, UK

[2] Barcelona Supercomputing Center, Barcelona, Spain

[3] LOCEAN, Sorbonne Universités

[4] National Oceanography Centre, European Way, Southampton, SO14 3ZH, UK

[5] Met Office Hadley Centre, Exeter, UK

[6] National Center for Atmospheric Research, Boulder, USA

*Correspondence to*: Pablo Ortega (pablo.ortega@bsc.es)

**Abstract.** The Subpolar North Atlantic (SPNA) is a region with prominent decadal variability that has experienced remarkable warming and cooling trends in the last few decades. These observed trends have been preceded by slow-paced increases and decreases in the Labrador Sea density (LSD), which are thought to be a precursor of large scale ocean circulation changes. This article analyses the inter-relationships between the LSD and the wider North Atlantic across an ensemble of coupled climate model simulations. In particular, it analyses the link between subsurface density and the deep boundary density, the Atlantic Meridional Overturning Circulation (AMOC), the Subpolar Gyre (SPG) circulation, and the upper ocean temperature in the eastern SPNA.

All simulations exhibit considerable multidecadal variability in the LSD and the ocean circulation indices, which are found to be interrelated. LSD is strongly linked with the strength of subpolar AMOC and gyre circulation, and is also linked with the subtropical AMOC, although the strength of this relationship is model dependent and affected by the inclusion of the Ekman component. The connectivity of LSD with the subtropics is found to be sensitive to different model features, including: the mean density stratification in the Labrador Sea; the strength and depth of the AMOC; and the depth at which the LSD propagates southward along the western boundary. Several of these quantities can also be computed from observations, and comparison with these observation-based quantities suggests that models representing a weaker link with the subtropical AMOC might be more realistic.

## 1. Introduction

The North Atlantic Ocean is a key component in Earth's climate through, for example, its role in redistributing heat and in taking up excess heat and carbon from the atmosphere. It is also a region that has varied significantly in the past. This is particularly true for the North Atlantic subpolar gyre, that has varied significantly on multi-decadal timescales across a range of different variables (Häkkinen and Rhines, 2004; Holliday et al., 2020; Reverdin, 2010; Robson et al., 2018b). Basin-mean sea surface temperature (SST) over the North Atlantic has also been observed to vary on multi-decadal timescales (Schlesinger and Ramankutty, 1994), and has been linked to a range of important climate impacts, including hurricane numbers and rainfall in monsoon regions (Knight et al., 2006; Monerie et al., 2019; Zhang and Delworth, 2006). The North Atlantic is also expected to change significantly in the future due to

the effects of climate change, and consequently produce substantial climate impacts on the surrounding regions
(Sutton and Hodson, 2005; Woollings et al., 2012). On decadal timescales, it is the interaction between natural
variability and externally forced changes that will shape how the Atlantic regions climate will evolve. Therefore, in
order to improve predictions of the North Atlantic, it is imperative that we improve our understanding of the
processes that control decadal timescale changes in this region.
It has generally been thought that changes in the ocean circulation, and particularly the Atlantic Meridional
Overturning Circulation (AMOC), have played a significant role in shaping the Atlantic Multidecadal Variability
(AMV; Knight et al. 2005). In particular, changes in the strength of the AMOC, and its related ocean heat transports
have been shown to control multi-decadal internal variability in a range of coupled climate models (Danabasoglu,
2008; Dong and Sutton, 2005; Jungclaus et al., 2005; Ortega et al., 2011, 2015). The proposed mechanisms to
explain the multi-decadal variability involve interplays between the North Atlantic Oscillation (NAO), North
Atlantic Deep Water (NADW) formation, the boundary currents, the Gulf Stream and gyre circulations, and the
horizontal density gradients (e.g. Joyce and Zhang, 2010; Polyakov et al., 2010; Ba et al., 2013; Nigam et al., 2018;
Zhang et al., 2019). Changes in AMOC and the wider ocean circulation have been indeed used to explain the
observed changes in the subpolar North Atlantic (SPNA) on decadal and longer timescales (Moat et al., 2019). In
particular, the SPNA underwent a rapid warming and salinification in the mid 1990s before a decadal timescale
cooling and freshening started in 2005, which is consistent with decadal-to-multidecadal variability of the AMOC
(Robson et al., 2012, 2013, 2016). The recent cooling has been linked to climate impacts over the continents,
including heat waves (Duchez et al., 2016), through an effect on the position on the jet stream (Josey et al., 2018). A
long term relative cooling of the SPNA since ~1850 has also been attributed to a centennial weakening of the
AMOC (Caesar et al., 2018; Rahmstorf et al., 2015), an AMOC reduction that most CMIP6 model projections
predict to continue in the future (Weijer et al., 2020). However, a lack of direct observations of the strength of the
AMOC or the ocean circulation more generally have hindered our ability to make a direct attribution of recent
changes.
In order to understand the aforementioned changes in the SPNA on multi-decadal timescales many authors have
turned to indirect measurements of the AMOC. One particular proxy of AMOC strength that has received some
focus recently are density anomalies at depth in the western SPNA or Labrador Sea region. In climate models,
density anomalies in the western SPNA are a key predictor of density anomalies further south on the western
boundary, and hence of the AMOC strength via thermal wind balance (Hodson and Sutton, 2012; Ortega et al.,
2017; Robson et al., 2014, 2016). Observations show considerable decadal variability in subsurface density
anomalies; density anomalies in the western SPNA or Labrador Sea between ~1000-2500 m increased significantly
and peaked in ~1995 and subsequently declined (Robson et al., 2016; Yashayaev and Loder, 2016). Therefore, these
density anomalies have been interpreted as indicating that the AMOC peaked circa mid-to-late 1990s, and then
declined, consistent with the warming and then cooling of the eastern SPNA (Hermanson et al., 2014; Ortega et al.,
2017; Robson et al., 2016). Time series of subsurface density anomalies in the western SPNA are also consistent
with other proxies of AMOC strength, including sea level based proxies (McCarthy et al., 2015; Sutton et al., 2018),
sediment based proxies (Thornalley et al., 2018), and upper ocean heat content fingerprints (Caesar et al., 2018;
Zhang, 2008). Furthermore, the decline in AMOC suggested by the above proxies is also consistent with the
observed AMOC decline at 26ºN since 2004 (Smeed et al., 2018), and also with the changes in AMOC seen in
ocean data assimilation systems (Jackson et al., 2016, 2019). Therefore, there is confidence that large scale changes
in North Atlantic Ocean circulation have occured over the past few decades, and that they have had a significant
impact on upper ocean heat content.
Although there is consistency across proxies of AMOC changes in the North Atlantic, there are considerable gaps in
our understanding and major uncertainties to overcome. For example, the development of the subsurface density
proxies has been investigated so far in just a few models (Ortega et al., 2017; Robson et al., 2014). However, there is

considerable spread across climate models in the simulations of AMOC mean state and variability (Reintges et al., 2017; Zhang and Wang, 2013), and also in the latitudinal coherence of AMOC anomalies (Li et al., 2019; Roberts et al., 2020; Hirschi et al., 2020), which might reflect different roles of deep density anomalies in the western SPNA on the AMOC, as well as different interplays between the subpolar and subtropical gyre contributions (Zou et al., 2020). Models also do not resolve realistically many key features of AMOC, most notably the overflows, and this affects the subsurface stratification downstream and on the western boundary (Zhang et al., 2011). There also remains significant uncertainty for other important processes. For example, it is not yet clear whether the recent changes in the SPNA are an ocean response to buoyancy forcing, or whether mechanical wind forcing has shaped the recent observed changes (Robson et al, 2016; Piecuch et al. 2017). Local surface fluxes are also likely to explain a significant proportion of the recent cooling (Josey et al, 2018). Subsurface density anomalies are not just a proxy for  the AMOC, but more generally for buoyancy forced (or thermohaline) circulation changes, including gyre changes (Ortega et al., 2017; Yeager, 2015). Finally, the AMOC variability is also thought to respond to local wind forcing on a range of timescales, especially at lower latitudes (Polo et al., 2014; Zhao and Johns, 2014), which could disrupt or "mask" the influence of subsurface density anomalies as they propagate further south.

There is also considerable uncertainty in how and where subsurface density anomalies are formed in the SPNA, and how they are related to the AMOC. In observations and models, most water transformation associated with the AMOC occurs within the SPNA, and particularly in the eastern SPNA (Desbruyères et al., 2019; Grist et al., 2014; Langehaug et al., 2012). However, subsurface density anomalies in the western SPNA on decadal timescales have often been linked with buoyancy forcing and changes in deep convection in the Labrador Sea or with changes in the volume of Labrador Sea Water production (Yashayaev and Loder, 2016; Yeager and Danabasoglu, 2014). Many studies have also reported that the basin-wide AMOC in ocean-only and coupled models is sensitive to heat flux or buoyancy forcing in the Labrador Sea (Kim et al., 2020; Ortega et al., 2011, 2017; Xu et al., 2019; Yeager and Danabasoglu, 2014). Indeed, idealised experiments have shown that persisting positive NAO phases can strengthen the AMOC by fostering deep water formation via increased surface cooling in the Labrador Sea, thus inducing changes in the zonal density gradient (Delworth and Zeng, 2016; Kim et al., 2020), and thermal wind responses. However, the real link between deep convection, deep water formation, and density anomalies at depth in the Labrador Sea is complex, and not fully understood (Katsman et al., 2018). Observations suggest that very little water transformation and deep water formation actually occurs in the Labrador Sea (Pickart and Spall 2007; Lozier et al., 2019). Indeed, recently it has been shown that the Labrador Sea (i.e. OSNAP-west) played very little role in the interannual variability so far observed across the whole OSNAP line (Lozier et al., 2019), with the Irminger Sea playing a more dominant role. The Irminger Sea is a region that in some models controls the AMOC and SPNA variability, and that is especially sensitive to advective processes (Ba et al., 2013) and Arctic overflows (Fröb et al., 2016). Moreover, ocean-only models appear to significantly overestimate the amount of deep water formed within the Labrador Sea, with likely implications for coupled models (Li et al., 2019). These inconsistencies raise the question of whether models are simulating the right relationships.

In this study we will address some of the above uncertainties by performing a multi-model analysis of the North Atlantic in coupled climate models. We focus on the question of how robust is the relationship between subsurface Labrador Sea density anomalies and the basin-wide Atlantic Ocean circulation on decadal timescales. We also assess the question of whether Labrador Sea density can robustly induce density changes over the western continental slope and generate a geostrophic response in the meridional circulation (Bingham and Hughes, 2009; Roussenov et al., 2008). Shedding new light on these links is important for, among other reasons, determining to what extent the RAPID measurements represent the variability of the basin-wide AMOC cell, as well as to identify the models that can produce more reliable predictions and projections of the SPNA. For this, we will assess specifically the connection between subsurface density and AMOC at high and low latitudes via the western boundary. Furthermore, we will determine whether models consistently support an impact of AMOC changes on the SPNA upper ocean temperatures, and if not, investigate why. Our primary aim is to provide, for the first time in a multi-model context,

a broad characterization of these relationships using consistent analysis frameworks and tools, and to document the
uncertainty. The reasons for the uncertainty in the relationships will also be explored, establishing links with key
model climatological properties that could eventually be exploited as emergent constraints. We intentionally do not
explore in detail how subsurface density anomalies are formed in these models, and leave this for further study.
The paper is organised as follows. Section 2 describes the experiments and methods. Labrador Sea density, and its
link with the ocean circulation and the wider North Atlantic are explored across the multi-model ensemble in
Section 3. The characteristics of the intermodel spread in the previous relationships are explored in Section 4, and
Section 5 presents the main conclusions of this study and discusses its implications.
**2.    Experiments and methods**
Here we provide an overview and brief description of the models used in this study and provide some statistical
considerations  for the intermodel comparison.
**2.1.    Experiment selection**
For the multi-model analysis, we use the preindustrial control simulations (picontrol) from the fifth phase of the
Coupled Model Intercomparison Project (CMIP5; Taylor et al. 2012), in which forcing values of GHGs, aerosols,
ozone and solar irradiance are fixed to 1850 levels. We chose to use control over historical simulations to focus
exclusively on internal variability and benefit from the more robust statistics that the long preindustrial experiments
provide. Furthermore, we avoid the forced trends present in the historical experiments, which can lead to
correlations that are difficult to interpret objectively (Tandon and Kushner, 2015). From the CMIP5 ensemble , we
only use those models in which 3D fields of ocean temperature and salinity, as well as the streamfunctions of
meridional overturning circulation and/or the barotropic circulation, were available. Twenty different models meet
this condition. Their main characteristics and number of simulation years have been summarized in Table 1. Most of
the models have a nominal horizontal resolution in the ocean close to 1°, and, therefore, cannot resolve the effects of
eddies. Menary et al. (2015) has shown for these same model simulations that the effective horizontal resolution can
be higher over the Labrador Sea, due to the non-regular grids. Effective resolutions over the Labrador Sea area range
from 0.21° in the GC2 model to 1.1° in GISS-E2-R/GISS-E2-R-CC/CanESM2, with these differences determining
to a large extent the mean state model biases and the dominant drivers (i.e. salinity or temperature) of the Labrador
Sea density changes.
Complementing these simulations, we also consider two control experiments with eddy-permitting resolutions.
Specifically, we use a present day control simulation (i.e. with fixed radiative forcing levels from year 1990) of the
HiGEM model, with nominal horizontal resolution in the ocean of 1/3°, and of 0.83° latitude × 1.25° longitude in
the atmosphere  (Shaffrey et al., 2009), and a pre-industrial control of HadGEM3-GC2 (hereafter, GC2; Ortega et al.
2017) with a nominal resolution in the ocean of 1/4° (ORCA025) and N216 in the atmosphere (i.e. approximately 60
km in the mid-latitudes). The GC2 simulation is the same one employed for the previous analyses of Labrador Sea
variability in Robson et al. (2016) and Ortega et al. (2017). Note that we will assume that the present day control in
HiGEM can be compared with the other preindustrial simulations due to the large uncertainty these later show in
their climatological biases, and so, for the sake of simplicity, we will only refer to preindustrial control experiments
from now on. Figure 1 demonstrates that this assumption is reasonable, since the mean Labrador Sea stratification in
HiGEM is very similar to that in the other models.
As an observationally-constrained reference, this study also includes the assimilation run from DePreSys3, a decadal
prediction system from the MetOffice based on GC2 (Dunstone et al., 2016). In the ocean, the assimilation is

performed through a strong nudging (ten-day relaxation timescale) towards the full fields of a three-dimensional objective temperature and salinity analysis (Smith and Murphy, 2007). Since it covers a comparatively shorter period (1960-2013), and therefore different timescales than the control experiments, its comparison with the other simulations will be done with caution, in particular regarding the indices of the large-scale Atlantic circulation, for which other assimilation products show important discrepancies (Karspeck et al., 2015), thus highlighting significant uncertainty. For evaluation purposes, we also use EN4.2.1 (Good et al., 2013), an objective analysis of monthly temperature and salinity 3D observations developed at the MetOffice.

**Table 1:** List of the models used for this study, their characteristics and those of their picontrol simulations. For further details on the CMIP5 model configurations and components please refer to Table 9.A.1 in Flato et al. (2013) and references therein.

| Model ID | Lon x Lat ocean resolution (number of vertical levels) | Length | Key variables available |
|---|---|---|---|
| HadGEM3-GC2 | 1/4° x 1/4° (75 levels) | 311 years | AMOC, SPGSI, LSD, NOHT |
| HiGEM | 1/3° x 1/3° (40 levels) | 341 years | AMOC, SPGSI, LSD, NOHT |
| ACCESS1-0 | 1° x 1° enhanced near Equator and high latitudes (50 levels) | 500 years | SPGSI, LSD, NOHT |
| ACCESS1-3 | 1° x 1° enhanced near Equator and high latitudes (50 levels) | 500 years | SPGSI, LSD, NOHT |
| CCSM4 | 1.125° x 0.27–0.64° (60 levels) | 1051 years | AMOC, SPGSI, LSD |
| CESM1-BGC | 1.125° x 0.27–0.64° (60 levels) | 500 years | AMOC, LSD |
| CESM1-CAM5 | 1.125° x 0.27–0.64° (60 levels) | 319 years | AMOC, LSD |
| CESM1-FASTCHEM | 1.125° x 0.27–0.64° (60 levels) | 222 years | AMOC, LSD |
| CESM1-WACCM | 1.125° x 0.27–0.64° (60 levels) | 200 years | AMOC, LSD |
| CNRM-CM5 | 0.7° x 0.7° (42 levels) | 850 years | AMOC, SPGSI, LSD |
| CanESM2 | 1.4° x 0.93° (40 levels) | 996 years | AMOC, SPGSI, LSD |
| FGOALS-g2 | 1° × 1° with 0.5° meridional in the tropical region (30 levels) | 700 years | AMOC, LSD |
| FGOALS-s2 | 1° × 1° with 0.5° meridional in the tropical region (30 levels) | 501 years | SPGSI, LSD, NOHT |
| GFDL-ESM2G | 1° × 0.85° (63 levels) | 500 years | SPGSI, LSD |
| GISS-E2-R | 1.25° x 1° (32 levels) | 550 years | AMOC, LSD |
| GISS-E2-R-CC | 1.25° x 1° (32 levels) | 251 years | AMOC, LSD |
| MPI-ESM-LR | 1.5° x 1.5° (40 levels) | 1000 years | AMOC, SPGSI, LSD |

| MPI-ESM-MR | 0.4° x 0.4° (40 levels) | 1000 years | AMOC, SPGSI, LSD |
|---|---|---|---|
| MPI-ESM-P | 1.5° x 1.5° (40 levels) | 1156 years | AMOC, SPGSI, LSD |
| MRI-CGCM3 | 1° × 0.5° (51 levels) | 500 years | AMOC, LSD, NOHT |
| NorESM1-M | 1.125° x 1.125° (53 levels) | 501 years | AMOC, SPGSI, LSD, NOHT |
| NorESM1-ME | 1.125° x 1.125° (53 levels) | 252 years | AMOC, SPGSI, LSD, NOHT |

## 2.2. Methodological considerations

Density values are computed from 3D salinity and potential temperature fields, using the International Equation of State of seawater (EOS-80), and are referred to the level of 2000 dbar ($\sigma_2$), to give a stronger emphasis to the deep water properties.

Statistical significance of correlation coefficients is assessed following a two-tailed Student's t-test that takes into account the series' autocorrelation to correct the sample size, reducing the degrees of freedom of a series to its effective value (Bretherton et al., 1999).

Because our goal is to provide further insight into the suggested relationships established from observed trends in the North Atlantic (e.g., Robson et al., 2016), all statistical analyses in this study exploring the relationships between variables and associated lags are based on 10-year running trends. This is analogous to the calculation of a typical 10-year running mean, but computing over each 10 year period a linear trend instead and keeping the slope value. Note also that our main results remain similar if decadal running means are applied instead (not shown), as both are alternative approaches to concentrate on the low-frequency variability. Running trends have also the particular advantage of not being sensitive to long-term drifts, which are still present (and can be important for some simulations and variables) when running means are computed. To illustrate how decadal running trends represent low-frequency variability, and how they compare with the decadal running means, both have been included in Figure 2b (solid thick lines vs dashed thin lines) for an index of Labrador Sea density.

## 3. Labrador Sea density as an index of multi-decadal North Atlantic variability

This section explores the potential of Labrador Sea density as a proxy of the ocean circulation changes in the North Atlantic. As in our previous studies (Ortega et al., 2017; Robson et al., 2016), the indices that we will herein define represent waters within the Labrador Sea and not those that are necessarily formed in the region (e.g. Labrador Sea Water). Since Labrador Sea variability is affected by different processes (e.g. vertical mixing, Arctic-Atlantic overflows, sea ice interactions) that can be represented differently in the models, both in time and space, we characterize its variability over a relatively broad box (60°W-35°W; 50°N-65°N, blue box Figure 1a) that also includes part of the Irminger Sea region. Note that over this large area EN4.2.1, shows the weakest density stratification in the North Atlantic (characterised in Figure 1a as the density difference between 1000m and the surface).

## 3.1. Labrador Sea density across models

A first indicator of potential model discrepancies is Labrador Sea stratification, which can lead to differences in the representation of deep ocean convection (i.e. weaker density stratifications will facilitate the mixing, fostering convection activity, and vice versa for stronger density stratifications). Figure 1b-d illustrates the inter-model

differences with the vertical profile of the spatially averaged Labrador Sea temperature, salinity and density. The largest discrepancies are seen for temperature. Most models present their warmest waters at the surface, and temperatures decrease sharply to minimum values around 100 m and increase again at deeper levels, reaching uniform conditions after approx 300 m. However, the location and magnitude of this temperature minimum and the two maxima are highly variable. It is important to note that the profile for one of the models, MRI-CGCM3, is noticeably different to the others, with a subsurface minimum more than 2 degrees colder than for any of the other models. In terms of salinity, the general profile is more coherent across models, with minimum salinity at the surface that progressively increases with depth and attains uniform values after 500 m. Density stratification seems to be determined by salinity, as their two vertical profiles show similar features. This similarity includes exceptionally strong density and salinity stratification in MRI-CGCM3 as compared with the other models. This stratification is so strong that it precludes the occurrence of deep convection (not shown). Because of this, MRI-CGCM3 is an outlier for many of the metrics used in the paper, and has been excluded for the subsequent analyses to facilitate the interpretation of our results. We also note that the profiles for the two eddy-permitting models (green and orange lines in Figure 1b,d) lie within the spread of the CMIP5 models, indicating that resolution (at least to eddy-permitting spatial scales) does not drastically change stratification in the region. The DePreSys3 assimilation run closely matches the stratification in EN4.2.1, which supports DePreSys assimilation run as a reasonable observation-constrained reference for the models. The comparison of both observation-based datasets with the rest of simulations suggests that, in the subsurface, all models are too warm and most of them are too salty, two biases that have a competing effect on the mean subsurface density. Because of these canceling effects, several models show a comparatively better representation of the subsurface densities when compared to EN4.2.1 and DePreSys3. This compensation of model shortcomings for temperature and salinity is clearly illustrated in HiGEM, which shows a remarkable agreement with EN4.2.1 below 500 m.

To represent the characteristic interannual variability of Labrador Sea densities (hereafter referred to as LSD for consistency with previous work), we perform an Empirical Orthogonal Function (EOF; Storch and Zwiers, 1999) analysis and extract the leading mode for the spatially averaged annual means of LSD (Figure 2a), as in Ortega et al. (2017). For all simulations the first EOF of LSD exhibits a vertical structure where density values are largest at or near the surface and gradually decrease with depth. Thus, this first EOF typically reflects situations in which the density stratification, as described by the climatological vertical profile in Figure 1d, is weakened or strengthened, which happens when the corresponding principal component takes positive and negative values, respectively. Some inter-model discrepancies are evident, in particular regarding the depths where the maximum density values are found, which can happen between the surface and 500 m. Despite these differences, the dominant timescales of LSD variability seem to coincide between models. For example, Figure 2b illustrates the first principal component of LSD (PC1-LSD) for GC2 and HiGEM, showing in both cases clear multidecadal variability. Furthermore, Figure 2c shows the Fourier spectrum analysis of the annual PC1-LSD values, and most models show enhanced PC1-LSD variability for periodicities between 5 and 30 years.

In addition to the PC1-LSD index we consider a deep LSD index as introduced in Robson et al (2016). The deep LSD index is defined as the 1000-2500 m vertical mean of the spatially averaged density over the same region as PC1-LSD. We now compare how both indices represent the low-frequency changes in LSD, described in this paper as decadal running trends. A lead-lag correlation between the decadal trends in both PC1-LSD and deep LSD indices shows that they are strongly correlated in all models. However, some differences emerge when considering the lag of maximum correlation (Figure 2d). This comparison might indicate, once again, that decadal variability of subsurface density is concentrated at different depths in different models. It is also possible that both indices are sensitive to changes in deep water formation in different locations (e.g. Irminger or GIN Seas), which could, hence, affect the depth and maximum lag of the correlations. Nevertheless, we adopt PC1-LSD for the rest of the analyses, as it has the advantage of adjusting in each model to the depths in which density variability is more prominent.

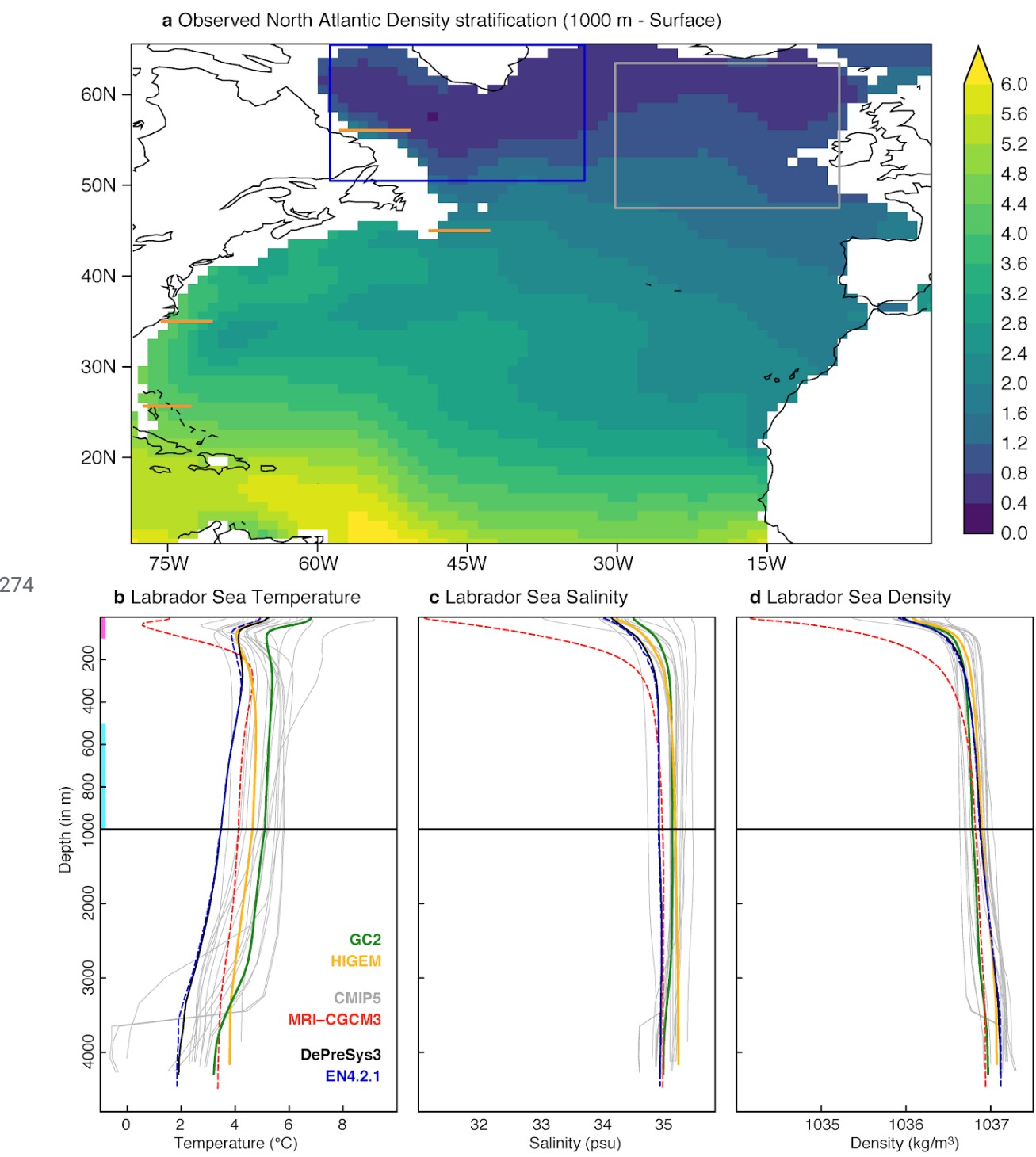

274

**Figure 1: a** Climatological density (computed as $\sigma_2$ at all depth levels) difference between the subsurface (1000m) and surface in the North Atlantic in the observational dataset EN4.2.1 (Good et al., 2013). The reference period to compute the climatology is 1960-2013. The grey box (32°W-10°W and 47°N-63°N) encloses the region where the ESPNA-T700 index in Figure 4d is computed. **b-d** Climatological mean of the spatially averaged Labrador Sea (60–35°W, 50–65°N, blue box in panel *a*) temperature, salinity and density as a function of depth in the simulation ensemble, the DePreSys3 assimilation run and EN4.2.1. The magenta (cyan) bars in the vertical axis correspond to

the depths that have been used to define the vertical stratification Labrador Sea indices. The horizontal orange lines by the North American coast represent the location of the latitudinal cross-sections in Figure 10 and Figure 11. For each model and dataset the climatology is computed for its whole length exceptv for EN4.2.1, that is computed for the overlap period with the DePreSys3 assimilation run.

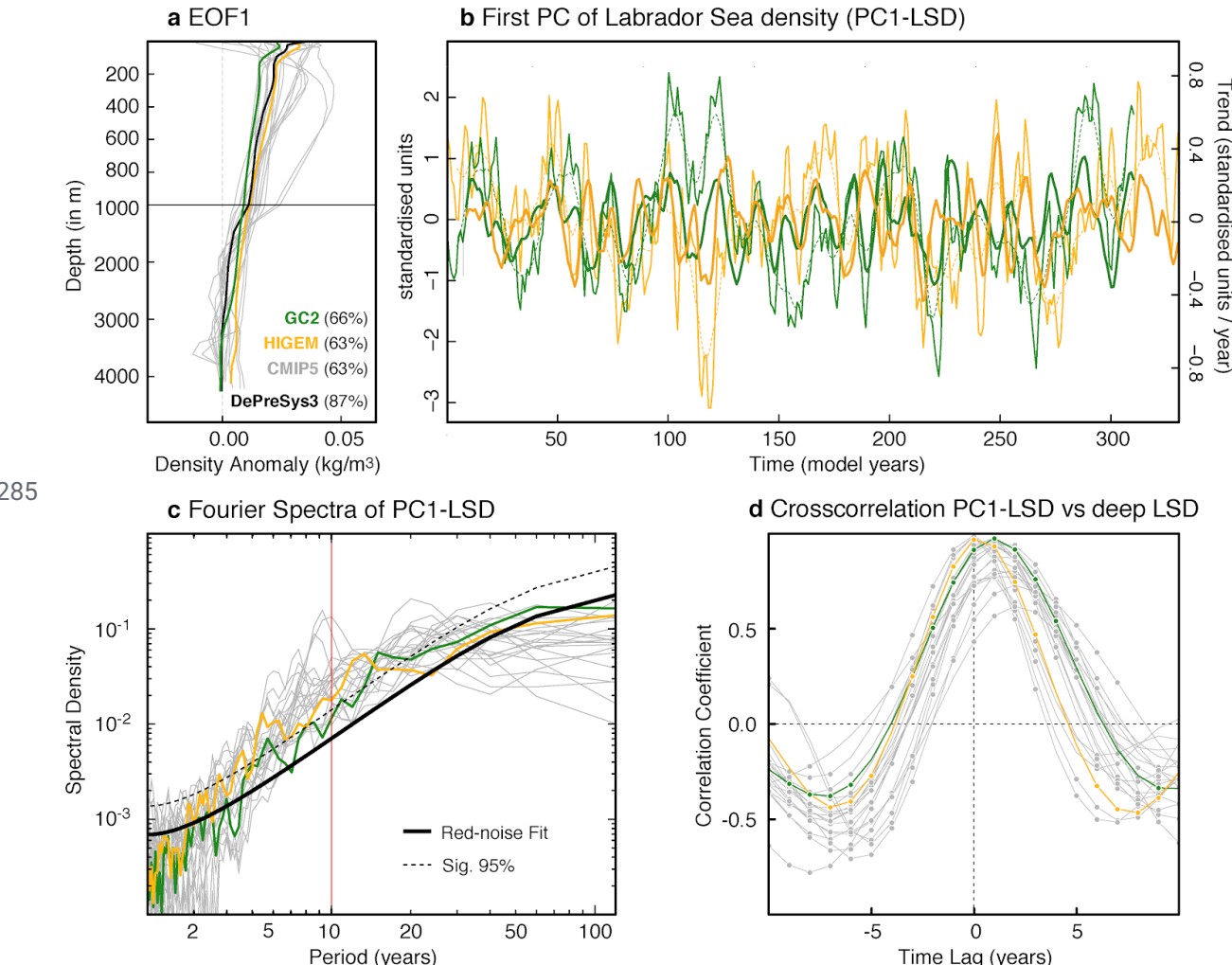

**Figure 2: a** First empirical orthogonal function (EOF) as a function of depth of the spatially averaged LSD in all the preindustrial experiments and in the DePreSys3 assimilation run. The percentage of variance explained by this mode in each model is included in brackets in the legend (for the CMIP5 runs, this represents the mean value across the ensemble). Because the sign of an EOF is arbitrary, it has been adjusted for all models (together with the sign of the respective principal component) so that both represent an increase in density stratification. **b** Associated principal component of the spatially averaged LSD (PC1-LSD) in the two high-resolution experiments. The thin solid lines represent the raw yearly-resolved PC1-LSD timeseries, the thin dashed lines their respective 10 year running means, and the thick (and slightly darker) lines their associated 10-year running trends (centered around the last year of the decade over which the trend is computed). **c** Normalized Fourier spectra of the PC1-LSD index in each of the preindustrial simulations. The black thick line represents a red noise process with the same first autoregressive (AR1) coefficient as PC1-LSD in GC2, and the dashed line sets the 95% confidence interval of this red-noise process. No major differences are found when using HiGEM's AR1 coefficient instead. The red vertical line highlights the 10 year periodicity to separate the interannual from the decadal to multi-decadal timescales. **d**

Lead-lag correlations between the decadal trends in PC1-LSD, and those in the deep LSD index from Robson et al.
(2016), defined as the 1,000–2,500 m average density in the box 60–35°W, 50–65°N. Positive lags indicate that
PC1-LSD leads the changes in deep LSD. Full dots denote correlation values exceeding a 95% confidence level
based on a student's t-test that takes into account the series autocorrelation.
**3.2.    Labrador Sea density linkages with the ocean circulation**
The link between PC1-LSD and other ocean circulation indices in the North Atlantic is now examined. Three
indices are considered: the AMOC at two different latitudes, 26°N (i.e. the same latitude as the RAPID array) and
45°N to capture the typical variability of the subpolar AMOC, and an index of the subpolar gyre strength. The
AMOC indices are computed as the maximum of the North Atlantic overturning circulation at any depth.
Furthermore, the Ekman component is removed to focus on the slow wind-forced and the thermohaline-driven (i.e.
the only one that can be influenced by the PC1-LSD directly) AMOC changes. To compute the Ekman component,
we vertically integrate the Ekman velocities (after introducing a depth-uniform return flow to ensure no net
meridional mass transport), following Eq. 6 in Baehr et al. (2004) with a fixed Ekman layer depth of 50 meters. This
Ekman component is then removed at each depth level, prior to the calculation of the AMOC indices. The subpolar
gyre strength is computed as an average of the North Atlantic barotropic streamfunction in the Labrador Sea region
(60–35°W, 50–65°N), where the gyre strength is usually maximum. Since the SPG circulation is cyclonic and,
therefore, associated with negative barotropic streamfunction values, the subpolar gyre strength index (SPGSI) is
multiplied by -1 so that an intensification of the gyre corresponds to a positive value of the index. The Fourier
spectra of the raw ocean circulation indices (Figure 3) show that, similar to the PC1-LSD, all three indices have
strong multidecadal variability, with the largest differences with respect to PC1-LSD emerging for the timescales
between 10 and 30 years, in which the spectral power is comparatively weaker, in particular for the AMOC26 index,
and at 50 and longer timescales, in which the ocean circulation indices appear to have enhanced variability with
respect to PC1-LSD. Similar spectra, but with enhanced variance at short timescales and reduced variance at the
longest timescales are obtained for the AMOC indices when the Ekman component is kept (Supplementary Figure
1), which suggests that the low-frequency processes dominate the total AMOC variability.
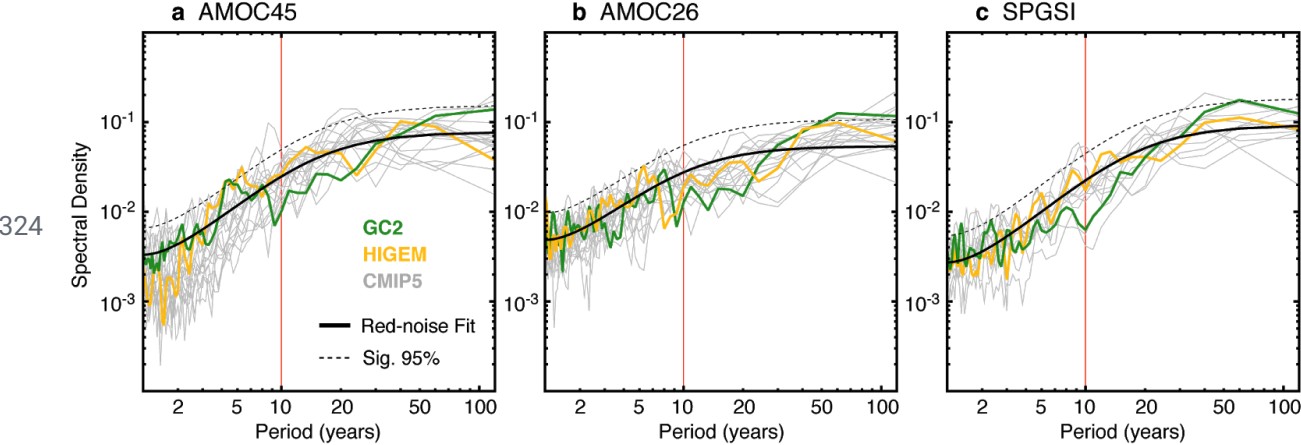
**Figure 3: a-c** Fourier spectra in the picontrol ensemble for the indices AMOC45, AMOC26 and SPGSI. Red-noise
spectra corresponding to a 1$^{st}$ order autoregressive process fit to GC2 indices are provided as reference.
Figure 4a shows that decadal trends in PC1-LSD are associated with trends in the AMOC at 45°N (AMOC45).
Nevertheless, there is some inter-model spread regarding the lag of maximum correlation, which ranges between 0
and 2 years (with PC1-LSD leading), although both variables are in phase for the majority of models. The AMOC at
26°N (AMOC26) is also positively related to PC1-LSD, with PC1-LSD leading AMOC26 by three years on average

(Figure 4b). However, the average correlation between PC1-LSD and AMOC26 is weaker, and the spread in the magnitude and lag of the maximum correlation is larger than for AMOC45. Therefore, it appears that the link with the subtropics is weaker than for 45°N and that AMOC coherence between subpolar latitudes and the subtropics in coupled models is model dependent. This weaker link of PC1-LSD with the subtropical AMOC is not surprising, as the LSD anomalies need to propagate a longer distance along the western boundary, allowing for model differences in the representation of ocean currents and gyres to impact the timing and magnitude of the maximum correlations. The reasons for the spread in the relationship between PC1-LSD and AMOC26 are explored in Section 4. A strong relationship is also found between PC1-LSD trends and those in SPGSI (Figure 4c), of similar order than for AMOC45. Thus, overall, PC1-LSD is a good proxy for the large-scale ocean circulation in the Subpolar North Atlantic, and can also be a precursor for a fraction of the AMOC variability in the subtropical Atlantic.

PC1-LSD is also a good precursor of the full AMOC variability (i.e. including the Ekman transport), although the wind-induced fluctuations associated with the Ekman component can introduce differences in the lags of the maximum AMOC vs PC1-LSD correlations (Supplementary Figure 2). This different lag can be explained by the fact that when the Ekman component is included, the AMOC contains a signal that is instantaneously driven by basin-scale surface wind anomalies (such as the NAO) that are, ultimately, also linked to the heat loss in the subpolar North Atlantic, which induces a delayed influence on the PC1-LSD (Ortega et al. 2017). Hence, including Ekman can lead to counterintuitive relationships in some models, in which the AMOC appears to lead the PC1-LSD changes. Also, in the particular case of GC2, the interference of the two signals (i.e. the subtropical Ekman and the delayed PC1-LSD) turns the correlations in Supplementary Figure 2d insignificant, masking out the real influence of PC1-LSD on the subtropics. For those reasons, and to ease the interpretation of the lagged-relationships, the rest of the analysis is exclusively focused on the AMOC indices without ekman.

The role of PC1-LSD as a precursor of the AMOC is further supported by a parallel analysis in Figure 5, looking at the maximum correlation between the decadal AMOC trends and those in Labrador Sea density as a function of depth, when the latter leads the AMOC by up to 10 years. Figure 5 reveals that the strongest link between the Labrador Sea densities and the AMOC, both at 45 and 26°N, occurs in its first 1000 m, the same levels where the first EOF of LSD show the maximum loadings (Figure 2a), which confirms the appropriateness of using PC1-LSD to represent the ocean circulation. The same analysis also supports a strong link between SPGSI and LSD, although in that case the largest correlations usually happen at deeper levels (between 1000 and 2000 m). Note also that the main conclusions drawn from PC1-LSD are also valid for the deep LSD index: however, the inter-model differences are larger in the cross-correlations with the AMOC indices (Supplementary Figure 3). This difference could reflect that the deep LSD index is more sensitive to other influences, like the Arctic overflows (Ortega et al., 2017), which can be very differently represented across models. Overall, the PC1-LSD index seems to be a better choice to describe multi-decadal North Atlantic variability in multi-model comparisons, as it selects the key depths for each model. However, PC1-LSD is mostly focused on near surface levels and, therefore, likely represents mostly Labrador Sea forced variability. Other indices describing densities at deeper levels might be preferable to compare Labrador Sea Waters of different origins across models, and to evaluate their realism against observations.

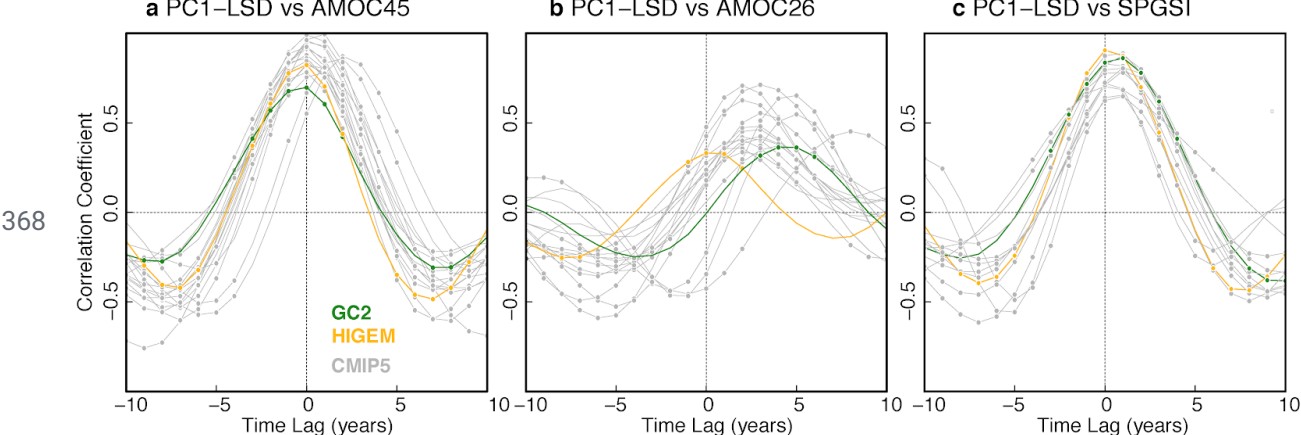

**Figure 4: a** Lead-lag correlations across the picontrol ensemble between the PC1-LSD index and the maximum AMOC streamfunction at 45°N after the Ekman transport is removed (AMOC45). Correlations are based on 10-year running trends. Significance is assessed as in Figure 2d and indicated with a circle. For positive lags, PC1-LSD leads. **b-c** The same as in *a* but between PC1-LSD and the maximum AMOC streamfunction at 26°N after the Ekman transport is removed (AMOC26) and the subpolar gyre strength index (SPGSI).

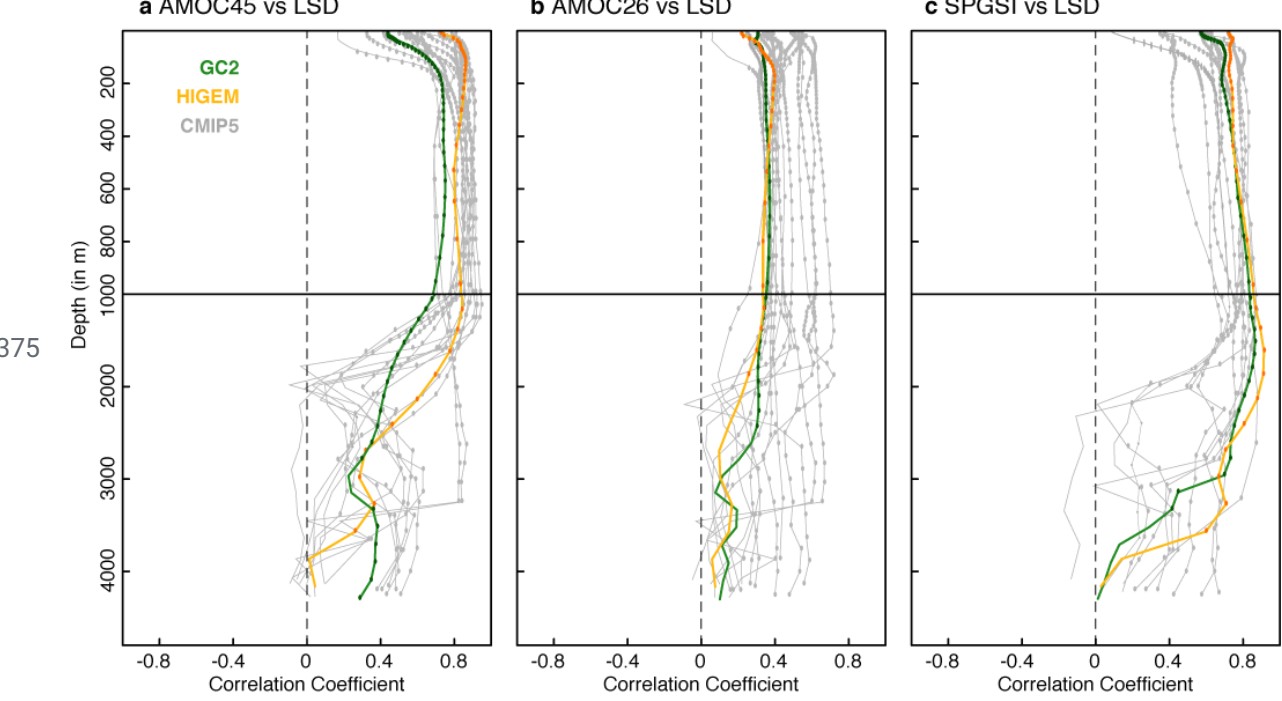

**Figure 5: a** Maximum correlation (for any lag between 0 and 10 years) between the AMOC45 index (after the Ekman transport is removed) and Labrador Sea Densities as a function of depth for all the simulations. Colored dots indicate correlations that are significant at the 95% confidence level. **b-c** The same as in *a* but between the AMOC26 index and LSD, and between the SPGSI and LSD, respectively.


### 3.3. Labrador Sea density linkages with the wider North Atlantic

Previous studies based on the GC2 picontrol simulation have suggested LSD to be also a potential predictor of
wide-spread cooling events in the eastern SPNA, like the observed cooling over 2005 to 2014 (Robson et al. 2016;
Ortega et al. 2017). We thereby continue our exploration of the PC1-LSD index by investigating its link with the
eastern SPNA in the multi-model ensemble. To explore this link we introduce a new index that represents the mean
potential temperature in the eastern SPNA region (32°W-10°W,47°N-63°N) averaged over the top 700 m of the
ocean (ESPNA-T700). Lead-lag correlations between the decadal trends in PC1-LSD and this index (Figure 6a)
show that there is a coherent relationship between both variables across models, with PC1-LSD increases
(decreases) being consistently followed by ESPNA-T700 warmings (coolings). Nevertheless, there are inter-model
differences concerning the magnitude and lag of the strongest positive correlations, revealing important uncertainty
in the relationship. The spread in the PC1-LSD vs ESPNA-T700 relationship is thus reminiscent of the spread found
between PC1-LSD and AMOC26, which suggests that they might be related. We also note significant negative
correlations when ESPNA-T700 leads PC1-LSD by 2-4 years that might be explained by the opposed (and nearly
concomitant) impacts that the NAO exerts on both variables (Figure 6b,c). Positive NAO phases, and associated
surface buoyancy forcing (Lozier et al., 2008) lead in first instance to negative SSTs (Barrier et al., 2014; Lohmann
et al., 2009) and an almost simultaneous cooling in ESPNA-T700 (Figure 6b). In comparison, on the western side
of the SPNA, positive NAO phases contribute to reduce vertical density stratification, favoring convection and a
more positive LSD index (Robson et al., 2016), which in the models lags the NAO by 2-3 years (Figure 6c). The
fact that correlations between NAO and ESPNA-T700 are weaker than between PC1-LSD and ESPNA-T700
suggests that the ocean might also be playing an additional role (besides the NAO) in controlling the ESPNA
temperatures.
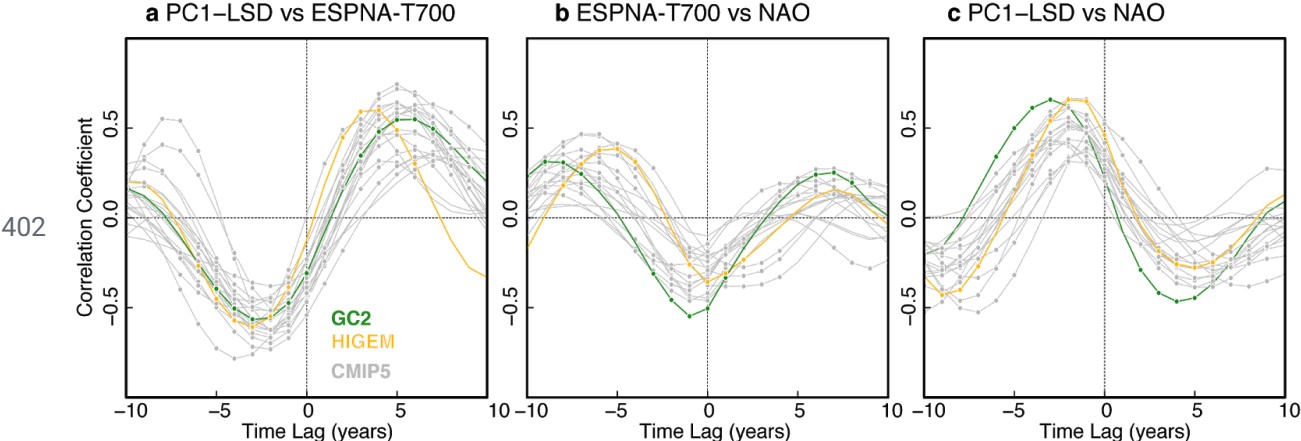
**Figure 6: a** Lead-lag correlations across the picontrol ensemble between the PC1-LSD index and the vertically
averaged top 700 m temperatures in the eastern subpolar gyre (ESPNA-T700; grey box in Figure 1a). Correlations
are based on 10-year running trends. Significance is assessed as in Figure 2d and indicated with a circle. For positive
lags, PC1-LSD leads. **b-c** The same as in *a* but between the North Atlantic Oscillation (NAO; defined as the
standardised difference in sea level pressure between the closest grid-points to Azores and Reykjavik) and the
ESPNA-T700, and between the NAO and the PC1-LSD, respectively. In these two cases, for negative lags the NAO
leads.

The link between PC1-LSD and the ESPNA could be explained through an influence of the PC1-LSD on the meridional ocean heat transport. This link is now investigated in the two eddy-permitting simulations (Figure 7) and in the five CMIP5 models for which the ocean heat transport fields are publicly available. In the two high resolution experiments and two of the CMIP5 ones the decadal trends in the meridional ocean heat transport at 45°N (OHT45) are strongly linked with those in PC1-LSD. This is a similar relationship to the one previously found in Figure 4 between PC1-LSD and both the AMOC45 and SPGSI, but in this case with PC1-LSD leading with slightly longer lead time. The other CMIP5 experiments support a weaker, yet significant, link, as well as a longer lag between OHT45 and PC1-LSD. Altogether, Figure 7a confirms that PC1-LSD is a good precursor of the changes in the meridional ocean heat transport, although with some differences across models which might reflect a different representation of certain processes. The contributions of two different processes to this delay are further investigated in HiGEM, for which OHT had been decomposed online at each time-step into vertical and horizontal heat transports (as in Bryan, 1969), which can be respectively interpreted as the "overturning" (i.e. characterised by the zonal mean transport) and "gyre" (i.e. characterised by variations from the zonal mean transport) components (Robson et al., 2018a). While the overturning contribution (OHT45$_{over}$) increases in phase with the AMOC45, SPGSI and PC1-LSD changes (Figure 7b), the increase in the gyre component (OHC$_{gyre}$) starts four years later. That lag could be the time required in HiGEM for the propagation of mean and/or anomalous temperatures from the southern to the northern branch of the SPG.

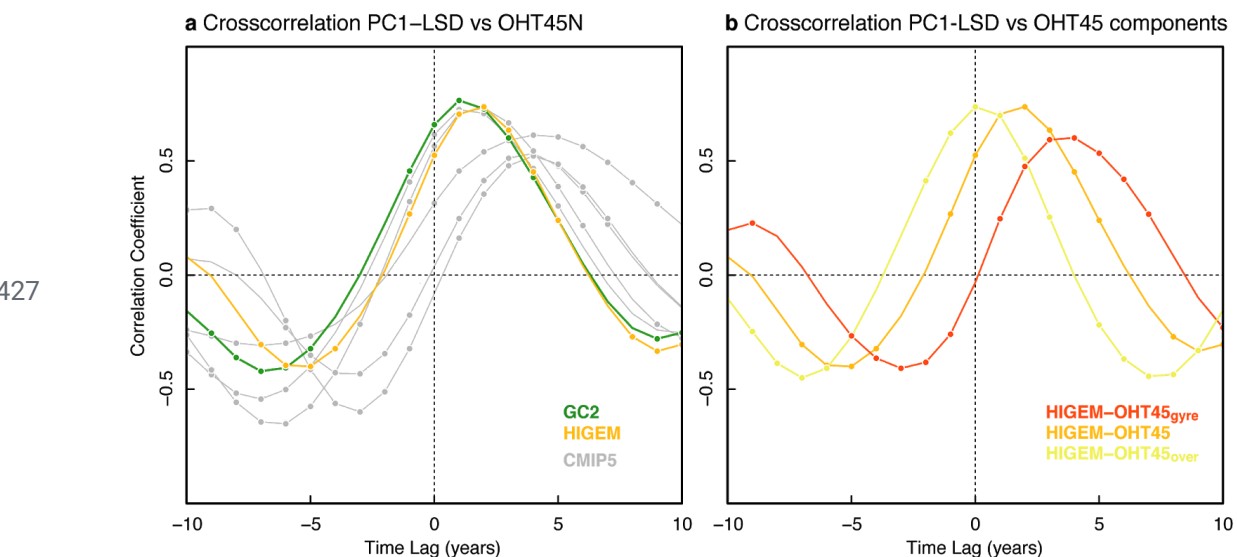

**Figure 7: a** Lead-lag correlations in a subset of the picontrol experiments between the PC1-LSD index and the ocean heat transport across the 45°N transect (OHT45N). Note that the ocean heat content is only available for 5 models of the CMIP5 ensemble. Correlations are based on 10-year running trends. **b** The same as in *a* but only in HiGEM for the different terms of the OHT45N. For positive lags, PC1-LSD leads.

### 4.    Characteristics of the inter-model spread in subpolar to subtropical AMOC

This section investigates which particular climatological model features are linked to the large inter-model spread in the PC1-LSD vs AMOC26 relationships. The most relevant model features thus identified will improve our process understanding, and can eventually be used to identify which models are most realistic and, in turn, can deliver more reliable projections of the future changes in the North Atlantic.

Figure 8 shows that models that simulate a stronger and deeper climatological AMOC (both at 45°N and 26°N) tend
to have a stronger correlation between PC1-LSD and the subtropics. All these linear relationships between
climatological AMOC strength and depth and the PC1-LSD vs AMOC26 connectivity are significant at the 95%
confidence level. These climatological AMOC values (without Ekman) can be put in context with those from
RAPID observations and DePreSys3. RAPID observational uncertainties have been considered by including the
mean values over three different non-overlapping periods (i.e. 2004-2007, 2008-2012 and 2013-2016; dotted lines in
Figure 8). The scatterplots show that the majority of models whose climatological AMOC26 lies within the
RAPID/DePreSys3 climatological spread have a relatively weak link between PC1-LSD and AMOC26, although
some models supporting a strong link are also included or remain close to the RAPID/DePreSys3 values. However,
caution is recommended, e.g., before defining emerging constraints, because model and observations are not directly
comparable for numerous reasons. For example, both RAPID and DePreSys3 cover shorter periods than the
simulations and relate to different background forcing conditions (present day vs preindustrial) which might imply
different mean states (Thornalley et al., 2018). Also, climatological values of the AMOC26 strength are notably
weaker in DePreSys3 than in RAPID, a difference that is not explained by the different temporal periods covered by
each dataset (not shown) and that implies that DePreSys3 might be underestimating too the real AMOC45 strength.
This underestimation might be larger than shown in Figure 8, as evidence suggests RAPID calculations from
mooring arrays might be underestimating the AMOC strength by ~1.5 Sv (Sinha et al., 2018).

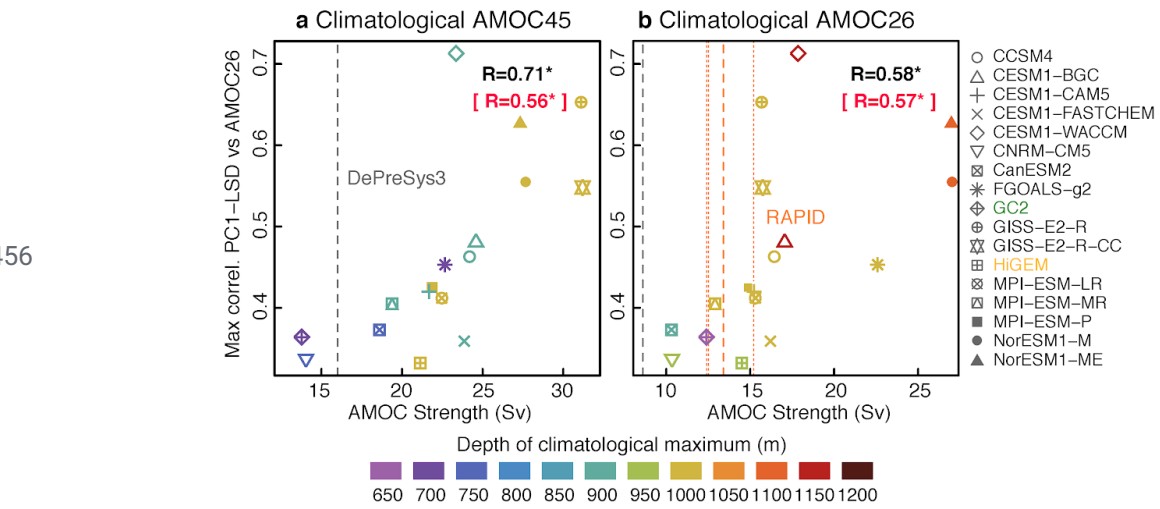

**Figure 8: a-b** Scatterplot of the maximum cross-correlation value in Figure 4b between PC1-LSD and AMOC26
against the climatological AMOC45 and AMOC26 means, respectively. All AMOC indices refer to the values after
the Ekman transport signal is removed. The maximum correlations are based on 10-year running trends, and always
happen when PC1-LSD leads the AMOC26 index. Colors indicate the depth at which the climatological AMOC
maximum occurs. The correlation coefficient between the maximum PC1-LSD correlation and the climatological
mean AMOC is shown in the top-left corner in black. In magenta, the analogous correlation but against the depth of
the mean climatological AMOC is shown. The presence of an asterisk indicates that the correlation is significant at
the 95% confidence level. The dashed grey vertical lines mark the climatological AMOC strength value in the
DePreSys3 assimilation run. The orange vertical lines indicate the climatological value from RAPID observations
(Smeed et al., 2018) from 2004 to 2016 (dashed), and in three non-overlapping sub-periods of 4 years (dotted).
A potentially important factor behind the inter-model spread in Figure 4b is the mean density stratification in the
Labrador Sea. Figure 9 suggests that, indeed, the PC1-LSD vs AMOC26 spread is partly influenced by the density
stratification in this region. Models that have a weaker density stratification (here defined as the difference between
the top 100 m, and the average between 500-1000 m), and thus favor deeper convection in the Labrador Sea,
generally exhibit a stronger link between PC1-LSD and AMOC26. This result is robust for other stratification
indices based on different depth levels (See Supplementary Figure 4). Differences in density stratification
across-models can be due to a combination of different factors, from differences in the local buoyancy fluxes (driven
by differences in the atmospheric circulation), to differences in the representation of the Arctic overflows, which are
parameterised in some models (e.g. the CESM family; Danabasoglu et al., 2010), and explicitly resolved in others.
No robust link between the PC1-LSD vs AMOC relationship and both temperature and salinity stratification in the
Labrador Sea has been found. It is also worth mentioning that all models except CanESM2 are more weakly
stratified in the Labrador Sea than the observations (represented herein by the DePreSys3 assimilation run and
EN4.2.1). Hence, the real link of LSD with the AMOC26 may not be as strong as some models suggest.

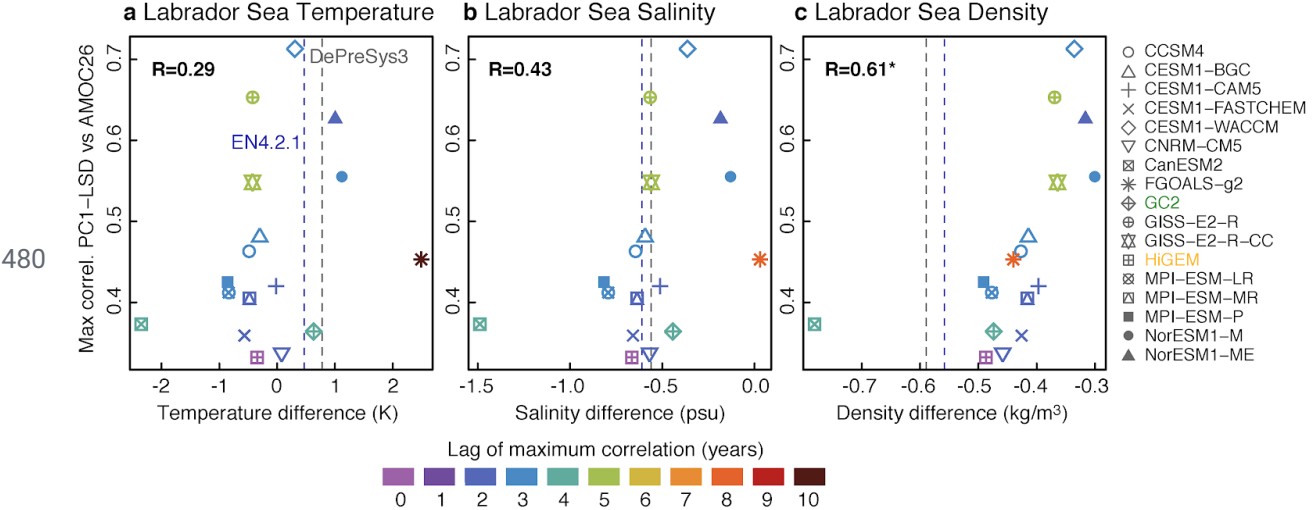

**Figure 9: a** Scatterplot of the maximum cross-correlation value in Figure 4b between PC1-LSD and AMOC26
(without the Ekman component) against the climatological mean of the Labrador Sea temperature stratification
index (computed as the difference of the vertical means in the levels 0-100 m minus the vertical means in the levels
500-1000 m; see Fig. 1). The maximum correlations are based on 10-year running trends. The correlation coefficient
between the two metrics is shown in the top-left corner. The presence of an asterisk indicates that the correlation is
significant at the 95% confidence level. Colors indicate the lag at which the maximum correlation between
PC1-LSD and AMOC26 is obtained. The grey (blue) vertical lines depict the mean stratification value in the
DePreSys3 assimilation run (EN4.2.1). In both cases, their overlap period is used to compute the climatology (i.e.,
1960-2013). **b-c** The same as in *a* but for the Labrador Sea salinity and density (defined as $\sigma_2$), respectively.
Another key aspect of the PC1-LSD vs AMOC26 connectivity is the western boundary density (WBD). Indeed,
boundary density is critical to the mechanism through which LSD influences the AMOC at lower latitudes. Positive
(negative) LSD anomalies propagate equatorward following this boundary, and as they do so they strengthen
(weaken) the zonal density gradient, triggering a thermal wind response that accelerates (decelerates) the AMOC. In
the following we investigate differences in the propagation of boundary densities across models, and if these
differences can affect the inter-model PC1-LSD vs AMOC26 spread. Figure 10 focuses on the two high-resolution
simulations, where important differences already manifest. It represents the in-phase correlations of PC1-LSD with
the density fields (defined as $\sigma_2$) near the western boundary at four different longitudinal transects: 57°N (cutting
across the Labrador Sea), 45°N, 35°N and 26°N. In both models, the depth of the maximum correlation near the
continental shelf is coherent across latitudes. However, in HiGEM these occur at deeper levels (1000 to 3000 m)
compared to GC2 (1000 to 2000 m), and the difference is especially clear at 35°N, where the highest correlations
occur at ~2000 m in HiGEM, while only at 1000 m in GC2. Similar depth differences are also found at 26°N, but
with slightly weaker correlations. In addition to the difference in the depth of the maximum correlation between
HiGEM and GC2, there are differences in the vertical structure between the two models. For example, at 35°N in
GC2, density anomalies on the western boundary form a tripole (low correlation above and below the maximum
correlation at ~1000 m), but in HiGEM the density anomalies form a dipole (Figure 10g). We note some differences
in bathymetry at this latitude (which is steeper in HiGEM), which might partly explain some of the differences in
terms of the density correlation structure.

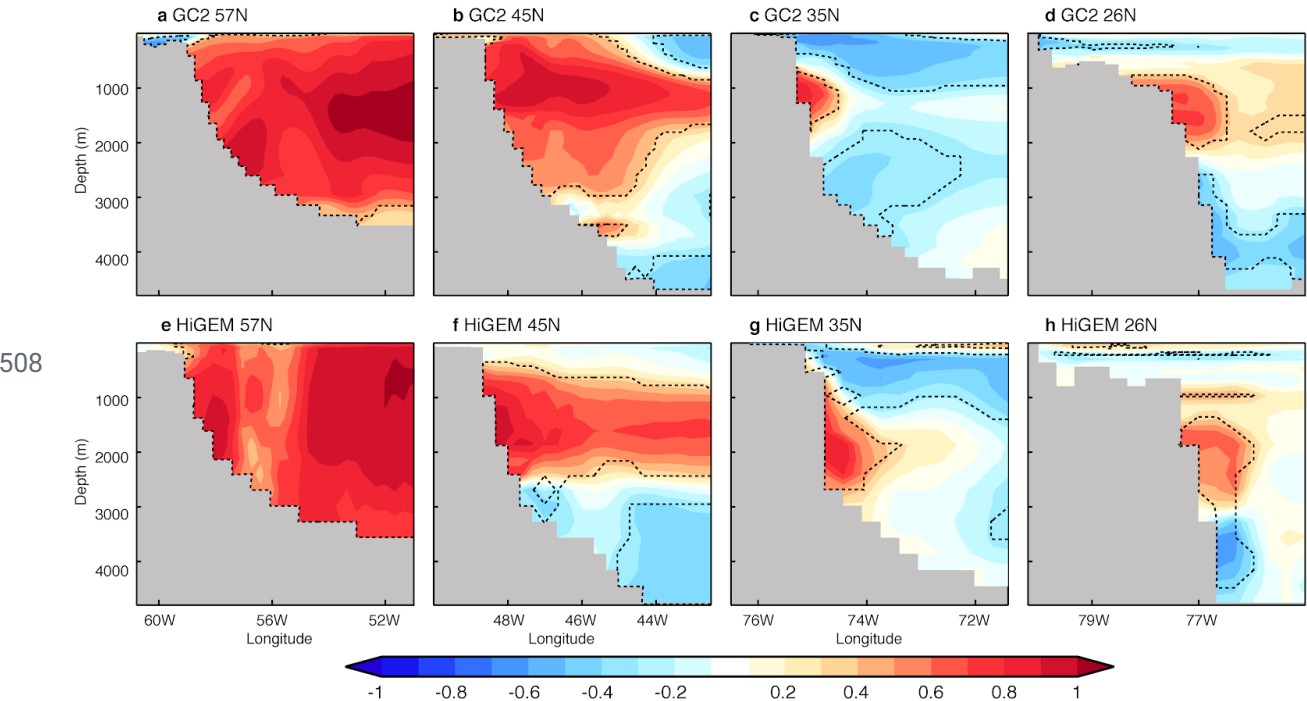

**Figure 10: a** In-phase correlation in GC2 between the PC1-LSD index and the density fields across a zonal section
at 57°N located in the vicinity of the western Atlantic boundary. Thin dashed contours enclose areas where the
correlation significance exceeds the 95% confidence level. Correlations are based on 10-year running trends. **b-d**
The same as in *a* but for zonal sections at 45°N, 35°N and 26°N. **d-h** The same as in *a-d* but for HiGEM.

Figure 11 shows that the diversity in the depth of these boundary densities is even more evident when including the
CMIP5 models. The depth of the maximum correlation between PC1-LSD and the western boundary density at the
four latitudinal sections relates linearly (and significantly at the 95% confidence level) across models with their
PC1-LSD vs AMOC26 correlation. In this case, models exhibiting maximum correlations with the WBDs at deeper
levels generally show stronger links between PC1-LSD and the subtropical AMOC. In DePreSys, our
observationally-constrained reference (dashed grey lines in Figure 11), these maximum correlations tend to occur at
relatively shallow levels when compared with the multi-model ensemble. We have also checked if models with
stronger correlations with the WBDs (as represented by the PC1-LSD and WBD maximum correlations at every
latitudinal section) also support a stronger link between the PC1-LSD and the AMOC , but this linearity assumption
only holds true at 57°N (correlations in magenta in Figure 11). This suggests that the depth along which WBDs
propagate southward, and/or the vertical structure of anomalies, are the key aspects to understand and potentially
narrow down the spread.

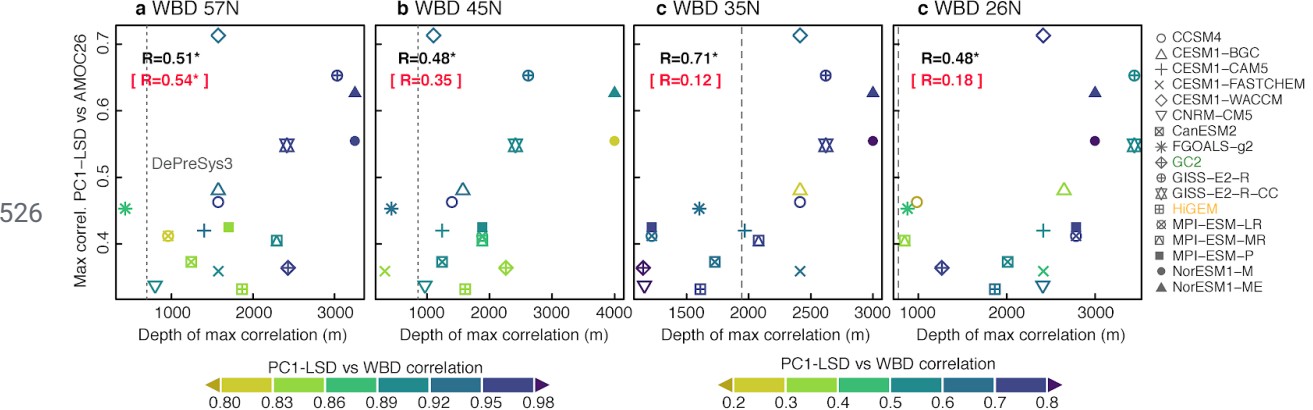

**Figure 11: a** Scatterplot of the maximum cross-correlations value in Figure 4b between PC1-LSD and AMOC26 (without the Ekman component) against the depth at which the maximum correlations at any lag between PC1-LSD and the WBD at 57°N occur. The maximum correlations are based on 10-year running trends. The correlation coefficient between the two metrics is shown in black the top-left corner. Likewise, another correlation coefficient in magenta is shown, computed between the PC1-LSD and AMOC26 maximum correlation and the PC1-LSD and WBD at 57°N maximum correlation. The presence of an asterisk indicates that the correlation is significant at the 95% confidence level. Colors indicate the maximum correlation between PC1-LSD and the WBD. The grey vertical lines depict the corresponding depth of maximum correlation for the DePreSys3 assimilation run. **b-d** The same as in *a* but for the WBD at 45, 35 and 26°N, respectively.

## 5.   Conclusions and discussion

This article has explored, in a multi-model context, the linkages between subsurface density in the subpolar North Atlantic (SPNA) and the ocean circulation further south. In particular, it has explored the role of Labrador Sea density (LSD) in driving Western Boundary Density anomalies (WBD) and the ocean circulation, and the impact on upper ocean temperature changes in the SPNA. The analysis was based on two control simulations with eddy-permitting models, a preindustrial one with HadGEM3-GC2 and a present day one with HiGEM, and on 20 CMIP5 preindustrial experiments. Furthermore, where possible these characteristic model features have been computed in observational datasets, as well as in a simulation assimilating observations. The major findings are listed below:

- All the simulations show clear multidecadal variability in Labrador Sea density. There is also a close link between LSD and the strength of the subpolar Atlantic Ocean circulation, with positive density anomalies leading to a strengthening of the Atlantic Meridional Overturning Circulation (AMOC) at 45°N and the Subpolar Gyre (SPG) circulation.
- The relationship between anomalous LSD and the strength of the AMOC at 26°N - the latitude of the RAPID array measurements - is also positive in the simulations, but there are significant inter-model differences, both in the strength of the relationship and the lag of maximum correlation. This uncertainty implies that the connectivity of LSD with the subtropics and latitudinal AMOC coherence is model-dependent.
- The connectivity between anomalies in LSD and AMOC at 26°N is sensitive to different model features, including the strength and depth of the climatological AMOC maximum, the mean density stratification in the Labrador Sea, and the depths at which the LSD propagates southward along the western boundary.

Stronger LSD connectivity with the subtropics tends to occur in models with a stronger and deeper AMOC, weaker Labrador Sea stratification and western boundary density propagating at deeper levels.

- Observationally derived constraints of the model based relationships tend to suggest that the link between LSD and the subtropical AMOC is weak. This suggests that observations of AMOC via RAPID may not be representative of the basin wide buoyancy forced AMOC variability. However, caution is advised because simulations and observations are not directly comparable, and so significant uncertainty remains in constraining the relationship between LSD and subtropical AMOC.

- The multi-model ensemble does also support a significant lagged relationship between LSD and the upper ocean temperature in the eastern SPNA, in line with previous studies linking LSD to the recently observed changes in the North Atlantic. However, models disagree regarding the strength of the link (correlations between 0.3-0.7), and the maximum lag (3 to 10 years).

We have shown that, in coupled climate models at least, subsurface density anomalies in the western SPNA are an important predictor of the wider North Atlantic ocean circulation and upper ocean temperature in the SPNA. This importance on the ocean circulation is especially clear at the latitudes of the SPNA itself. Given the important role of the wind in driving lower latitude AMOC anomalies, and the range of processes by which wind can act on the AMOC (Duchez et al., 2014b, 2014a; Kanzow et al., 2010; Polo et al., 2014; Zhao and Johns, 2014) it is not surprising that the relationship between LSD and AMOC at 26°N is much weaker. Nevertheless, the reasons behind the large spread in these relationships across models is not so clear.

We have tried to constrain this uncertainty by looking at a range of observed metrics that may explain the spread in the correlation strength, including the density anomalies on the western boundary, the stratification of the Labrador Sea, and the mean-AMOC strength. Overall, these constraints point to a relatively weak relationship between LSD and AMOC at 26°N on decadal timescales (i.e. r ~ 0.4) in the real world. However, there are many reasons why this number is still very uncertain, and further work is needed to assess its validity. A caveat of this study is that the simulations and observation-based datasets employed are not directly comparable, as they differ in the background radiative forcing levels, the length of the period used to compute the climatologies, and even the way some indices, like the AMOC, are computed. We also recognise that there is large uncertainty within the observationally derived metrics. For instance, the assimilation run in DePreSys3, which is used to constrain relationships, clearly underestimates the mean AMOC strength at 26°N with respect to RAPID (see Figure 8b) and, therefore, might be also underestimating the AMOC at higher latitudes. Our findings might also be limited by model deficiencies. There is emerging evidence that current models underestimate AMOC and North Atlantic variability on decadal timescales (Roberts et al., 2013; Cheung et al., 2017;), which can degrade decadal predictability in the region and even lead to overly weak linkages between the AMOC and the AMV (Yan et al., 2018). The AMV is indeed a mode of variability that also shows important differences across models, in different aspects like its periodicity, amplitude, spatial structure and climate footprints (Medhaug and Furevik, 2011; Zhang and Wang 2013; Kavvada et al., 2013), inter-model differences that could be partly connected with those herein reported for the PC1-LSD vs AMOC relationships. Models also tend to generally underestimate the depth of the return flow, and this may still affect how density anomalies project on the basin-wide AMOC. It has also been argued that ocean-only models produce too much deep water in the western basin and Labrador Sea (i.e., Li et al., 2019), and recent observations even challenge the prevailing view from models that Labrador Sea convection dominates the AMOC variability (Koenigk and Brodeau, 2017), suggesting that the key deep water formation occurs in the Irminger Sea, a few hundred kilometers north east of the Labrador Sea (Lozier et al., 2019). Therefore, further in-depth study is warranted to narrow down the uncertainty in the real AMOC and PC1-LSD relationship.

Most of the models considered in this study have relatively coarse resolution, including non-eddying oceans (≥ 1°x1°), which means that they might be missing some key dynamics for the AMOC (Johnson et al., 2019) that could be important to represent realistic linkages. The current analysis also includes two models at eddy-permitting

resolution (HadGEM3-GC2 and HiGEM), whose relationships lie within the spread of those in the coarser models.
However, it could be that higher resolution is needed (e.g. enabling meso-scale eddies in subpolar latitudes) to
identify substantial differences (Hirschi et al., 2020; Johnson et al., 2019). A recent analysis based on
HadGEM3-GC3.1 (a later version of HadGEM3-GC2) configured at different horizontal resolutions has shown that
long-standing model biases affecting the North Atlantic are reduced at eddy-resolving resolution (1/12° x 1/12° in
the ocean), and that the strength of the AMOC, the boundary currents and the northward heat transport is higher than
for the coarser resolutions (Hirschi et al., 2020; Roberts et al., 2019). High resolution coupled models also generally
support the new view from OSNAP observations in which the largest fraction of AMOC variability (on sub annual
to decadal timescales) originates at the eastern SPNA (Hirschi et al., 2020). Eddy-resolving resolutions have also
been shown in a multi-model study (Roberts et al., 2020) to represent the AMOC response at 26°N differently in
future projections, leading to stronger declines than in non-eddying simulations, declines mostly associated with a
weakening in the Florida Current. Roberts et al. (2020) also compares the meridional coherence of the AMOC,
which does not seem to be resolution-dependent, a result that is in line with another multi-model comparison
between non-eddying and eddy-permitting simulations (Li et al., 2019).
Despite the current limitations in the models considered for this study, it is important to highlight that they provide a
rather consistent picture of a chain of relationships in the North Atlantic that is able to explain some of the recent
observed trends (Robson et al. 2016). This paper has broadly characterized this behaviour, and highlighted the
uncertainty. These relationships are also consistent with the mechanisms proposed by Yeager and Robson (2017) to
explain high levels of predictive skill in the SPNA on decadal timescales. Our analysis has also helped to identify
specific metrics (such as LSD stratification and the depth of the boundary density) that could be used as emergent
constraints for future projections, i.e. to subset the simulations expected to more realistically represent the future
changes in the region. Having a more realistic subpolar gyre stratification at present day conditions has been shown
in CMIP5 simulations to increase the probability of a future collapse in convection (Sgubin et al., 2017), that would
lead to a widespread SPG cooling. It remains to be tested if similar conclusions can be drawn from eddy-resolving
simulations.
**Code availability.** The main scripts used in the analysis and other supporting information that may be useful to
reproduce the results of this article are archived at the Barcelona Supercomputing Center and will be shared upon
request by the corresponding author.
**Data availability**. Outputs from the CMIP5 simulations can be downloaded from the corresponding ESGF node:
https://esgf-node.llnl.gov/projects/cmip5/. EN4 observations used in this study correspond to version 2.1 of the
dataset, available at https://www.metoffice.gov.uk/hadobs/en4/download-en4-2-1.html. Outputs from the GC2,
HiGEM and DePreSys3 simulations are available upon request to the corresponding author.
**Author contributions**. P. O., J. R. and R. S. conceived the study, which was later discussed and refined with the
other co-authors. M. M. downloaded and processed the CMIP5 data, computing the main climate indices. P. O. led
the analysis, and together with J. R. prepared the manuscript with contributions from all co-authors.
**Competing interests**. The authors declare that they have no conflict of interest.
**Acknowledgements**. We thank the UK Met Office for providing the model data of GC2 used in this study, and all
the research centers that contributed to CMIP5 and made their data available. This work was largely supported by
the NERC Projects "Dynamics and Predictability of the Atlantic Meridional Overturning and Climate Project"
(DYNAMOC, NE/ M005127/1) and "Wider Impacts of Subpolar North Atlantic Decadal Variability on the Ocean
and Atmosphere" (WISHBONE, NE/T013516/1). P. O. work was additionally supported by the Spanish Ministry of
Economy, Industry and Competitivity through the Ramon y Cajal grant RYC-2017-22772. J. R. was additionally
supported by the NERC ACSIS program and R. S. by NERC via the National Centre for Atmospheric Science
(NCAS).

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
