# Peer review of "multi-decadal variability in the North Atlantic: a multi-model"

_Earth System Dynamics, 2020_

## Referee Comment (RC1) · Anonymous Referee #1 · 7 Dec 2020

**1   General comments**

Ortega et al. analyze the internal multidecadal variability of the CMIP5 piControl ensemble ($\sim 1°$ horizonal resolution) and two additional model simulations with a higher resolution ($1/3°$ and $1/4°$). Cross-correlations show significant links between subsurface density variability in the western subpolar North Atlantic (represented by the principal component of density; PC1-LSD) and the AMOC at different latitudes when the Ekman component is removed. The correlation is not coherent over different latitudes and the driving factors of the relationship are derived.

[Figure]

The manuscript provides a new and certainly helpful index for density anomalies in high latitudes and the multi-model perspective offers robust insights into the internal variability of coupled climate models. The text is well written and good to follow. I am not very surprised by most of the findings as in the context of Atlantic Multidecadal Variability (AMV) analyses of various models showed similar results (see point 3 in the next section). The discussion of these is almost missing. Those and other issues are described in detail below. Once these are addressed, I would like to see the manuscript published as it improves our ability to "predict earth system change" (from the scope text of ESD).

**2  Specific comments**

The main rationale of this paper is the interpretation of cross-correlations of processes that exhibit a red frequency spectrum. Large significant correlations are found between (subsurface) density anomalies/variability in the western subpolar North Atlantic (spatial average over the Irminger Sea, Labrador Sea and the boundary current along Greenland; blue box in Fig. 1a) and the subpolar gyre circulation strength in the same area (SPGSI) and the AMOC at $45°$N (AMOC45). This link decreases with distance as shown for the subtropical large-scale circulation, represented by the AMOC at $26°$N (AMOC26).

I am not very surprised by these results: 1) The active tracers temperature and salinity are transported to the subpolar gyre (SPG) by the strong Gulf Stream and the North Atlantic Current. Both tracers change density and hence the circulation. The stronger the circulation, the stronger the tracer transport/input. In this very general sense, Fig. S1 e and f are nearly identical: here, it does not matter if one correlate the SPG strength with the deep Labrador Sea density itself (dLSD) or with its first principal component (PC1-LSD). Then, by definition, the rather red frequency spectrum yields smaller correlations with increasing lag and/or space. Based on this I would expect lower correlations with the remote AMOC26 compared to AMOC45. Density anomalies propagate with the Western Boundary Current (WBC) – or the western boundary densities (WBD) as used here – but model differences in re-circulations, current strength and location or diffusion yield a larger model spread on the long way to the subtropics.

2) Since you remove the Ekman component from the AMOC indices, short-term variability is removed. I speculate that if the Ekman component would be included, the spectra would be closer to white noise (e.g. Figure 2 d of Ba et al. 2013) and the correlations would decrease more rapidly due to the more short-term wind-induced fluctuations (Wunsch and Heimbach 2012). If this is the case, does the main finding, the multidecadal variability in the Labrador Sea density as a precursor of the AMOC, applies also in the real world (or at least in the models) with Ekman contribution? If not, what do we learn instead? The role of the Ekman component should already be mentioned in the abstract.

3) Similar findings were described in the context of the Atlantic Multidecadal Variability (AMV). As you refer to Knight et al. 2005 in the introduction, I think the findings should be discussed with those studies that relate local (subsurface) hydrography and large-scale ocean circulation variability (e.g. Polyakov et al. 2010, Ba et al. 2013, or preferably with section 3 of the review of Zhang et al. 2019). This is of particular interest since the CMIP5 ensemble tends to underestimate internal variability (Cheung et al. 2017).

In addition, I miss several points in the discussion/conclusions:

- The box chosen for PC1-LSD is rather large. From my experience, the signal will be dominated by the WBC and is possibly quite different from the Irminger or Labrador Sea interiors. Due to this choice, it is difficult to address some of the uncertainties that you summarize in the introduction (L93f). Wouldn't this study be good opportunity to test different locations by calculating e.g. PC1-IS for the

Irminger Sea?

- Likewise, it would be great if you could redo the calculations with the historical CMIP5 ensemble in addition to the pure internal variability of the picontrol runs. I am aware that this would be beyond the scope of the paper as it is at the moment, however, if you and your colleagues discuss the work further, I would really like to see these runs included.

- I miss a final statement if and why PC1-LSD is better suited than e.g. dLSD or any other measure for deep water formation activity? What kind of index should be used by the community? Can you give a recommendation?

**3 Technical corrections**

- Please state the version of EN4 you used somewhere in the main text (not only in the url in "Data availability").

- Regarding all scatter plots: it is a bit difficult for me to distinguish the yellow/light green symbols representing the different models (I need to zoom in with the pdf viewer, don't know how it looks printed). If you want to stick to the colors, you could use capital letters instead of the symbols?

- The references are very annoying to check due to missing line breaks.

L134f & L139f: Can you give the number of model years of the picontrol and present day simulations?

L141: Rephrase "IPSL" so that one can find the model in Table 1.

L149f: Can you briefly describe the main differences between picontrol and present day experiments) one sentence enough)?

L168: EOS-80 is deprecated. TEOS-10 should be used in the future.

L179f: I would like to see the 10-year running mean time series in addition to the running trends in Fig. 2b.

L211f: Most models are too dense by > 0.1 kg m$^{-3}$ at the surface and > 0.2 kg m$^{-3}$ at the subsurface as shown in many previous papers. In my view, "good representation" is too optimistic.

L215f: Please give a reference on how you calculated the EOF (there is also no ref in Ortega et al. 2017).

L218f: It's not clear what it means when the "vertical profile in Figure 1 is weakened or strengthened". Can you clarify this related to stratification? Does this apply to the temp/salt profiles as well?

L223: In what way is the multidecadal variability seen in Fig. 2b "important"?

L226: This sentence confuses since the deep LSD was not mentioned before. Why not something like "In addition to the PC1-LSD index we define the deep LSD (dLSD) as . . ."?

L237f: Add to the caption that $\sigma_2$ density is used at depth. Is the sigma symbol missing in "( 2)"? Can you add axes to Fig. 1a?

L245f: Is it possible to add calendar years e.g. as an upper x-axis to Fig. 2b since one of the runs represent present day? Is there a reason why you don't show a PC1-LSD time series of one (or more) of the CMIP5 models in Fig. 2b?

L266: Please state the depth d that you use as the Ekman depth since Baehr et al. 2004 give a range (50-100 m).

L271f: By chance, could you name the main differences of the Fourier spectra if the Ekman is included (one sentence enough)?

L287: AMOC45 and AMOC26 and not AMOC45N and AMOC26N.

L280f: Please also write at least one sentence to Fig. 4d (or remove Fig. 4d since Fig. 4d = Fig. 6a).

L292: You write that the cross-correlation lags in Fig. 5 are up to 10 years. In the caption of Fig. 5 you write that the lags can be any (for me, this indicates that lags > 10 years are possible). Can you please clarify that in the text and/or caption?

L296f: Can you speculate why the model spread increases when dLSD is used compared to PC1-LSD? Also, in my view, the model spread increases only for the AMOC indices. The cross-correlations with the SPGSI and ESPNA-T700 indices are more or less independent of the choice of LS density index (Fig. S1).

L312: "... 2005 to 2015 (Robson et al. 2016; Ortega et al. 2017)" → "... 2005 to 2014 (Robson et al. 2016)"

L314f: Put the ESPNA-T700 definition to section 3.2, where you describe Fig. 4 (Fig. 4d = Fig. 6a).

L314: "temperature" → "potential temperature"

L326f: The ocean plays a role in what? This sentence is a little strange to me (of course, the ocean plays a role?).

L336: Correct "of the first on the".

L377f: Do you take the maximum cross-correlations of PC1-LSD and AMOC26 or the maximum of the absolute values of the cross-correlations? It is difficult to judge from Fig. 4b but maybe you can clarify that all the maximum cross-correlations occur when PC1-LSD leads AMOC26 (if this is the case).

L378: AMOC45 and AMOC26 and not AMOC45N and AMOC26N.

L389f: The sentences "Models with ..." and "Models that have ..." have the same
content. Remove one of them.

L393(1): In Fig. 9, depth averages from 0-100 vs 500-1000 m were used. In the header of Fig. S2 b as well as in the caption of Fig. S2 is written that depth averages from 0-100 vs 500-1000 m were used. I am confused since these are the same depth ranges but for you derive different results in Fig. 9 c compared to Fig. S2 b.

L393(2): I am not totally convinced by the stratification definition used in Fig. 9 and Fig. S2. From Fig. 9 a and b you infer that "no link has been found between the PC1-LSD vs AMOC26 relationship and temperature and salinity stratification in the Labrador Sea" (L390). Assuming that you indeed find significant and non-significant correlations for the same variable depending on the depth range, I would rephrase the sentence from "no T/S link" to something like "weaker T/S link than density ... depending on depth range".

L403: green → blue; EN4 → (EN4)

L405: Add to the caption that $\sigma_2$ density is used at depth.

L407: LSD → PC1-LSD; Add " (WBD)" at the end of the sentence.

L412: "It represents the correlations" → "It represents the in-phase correlations" (or lag-0 or . . .)

L413f: Please add the section lines to Fig. 1a and state that $\sigma_2$ density is used.

L421: "Figure 10f" → "Figure 10g"

L426: "HiGEM 57" → "HiGEM 57N" in header of Fig. 10e.

L434: I find the formulation "deeper WBDs" a little misleading since its the correlation between PC1-LSD and the zonal density field which is deeper in those models, not the WBD itself (in the sense of a boundary current).

L435: The sentence "We also checked . . ." can only be understood together with the
caption of Fig. 11. Please clarify.

L440 & L444: "AMOC26N" → "AMOC26"

L444: "57N" → "57°N"

L446: "run ." → "run." and "WDB" → "WBD"

L462f: The second and third findings can be merged.

L481: "wstern"

L505: Given the high latitudes I would not say that two 1/3° and 1/4° horizontal resolution ocean models with a similar climatology as the 1° CMIP5 ensemble imply that the representation of the mesoscale might not be as important as the climatology itself. The first baroclinic Rossby radius is ∼10 km or lower and as you give with the 1/12° reference, resolution does matter.

L510: Is it really "including in" or just "including"?

Caption of Fig. S1: ". ." → ". "

**4  References**

Ba et al. 2013: https://link.springer.com/article/10.1007%2Fs00382-012-1633-4

Baehr et al. 2004: http://hdl.handle.net/11858/00-001M-0000-0012-006A-A

Cheung et al. 2017: https://journals.ametsoc.org/view/journals/clim/30/12/jcli-d-16-0712.1.xml?tab_body=fulltext-display

Polyakov et al. 2010: https://link.springer.com/article/10.1007%2Fs00382-008-0522-3

Wunsch and Heimbach 2012: https://journals.ametsoc.org/jcli/article/26/18/7167/34385

Zhang et al. 2019: https://agupubs.onlinelibrary.wiley.com/doi/full/10.1029/2019RG000644

---

## Referee Comment (RC2) · Anonymous Referee #2 · 4 Jan 2021

Variations of the Atlantic Meridional Overturning Circulation (AMOC) and the associated meridional heat and fresh water transports are an important driver of climate variability in the North Atlantic region and beyond. They are also thought to provide a source of predictability on decadal time scales. Previous studies have described mechanisms behind this variability often focusing on the source regions of North Atlantic Deep Waters. The authors describe the AMOC characteristics of about 20 CMIP5 models, most of them at resolutions in the ocean models of 0.5 to 1 degree, and two coupled models using an eddy-permitting ocean grid. They relate AMOC variability in the sub-polar and sub-tropical Atlantic to density changes in the Labrador Sea (LS). Such a connection has been invoked in several earlier studies, but the merit of this

manuscript is that they provide an investigation in a multi-model context. They concentrate on the buoyancy-driven part of the overturning circulation and use long pre-industrial control simulations to provide statistically sound results. A striking result is that there is a high degree of coherence in the models (at least in this specific class of models) regarding the relation between the LS density structure and the sub-polar North Atlantic circulation. Even the corresponding relation to the lower latitudes is still quite robust, even though model differences lead to different strengths of the correlations. The study itself and the individual analyses are well thought-through and the text describes adequately the results presented in the figures. Apart for sharing the critique of the first reviewer on missing discussions of some important previous papers (I would also include, for example, Delworth and Zeng (2016) where the mechanisms are nicely described), I feel that the authors overstate a bit the robustness of the "observational constraints". For example, given the results of Jackson et al. (2019), I don't think that the numbers of AMOC strength from DePreSys (used in figure 8) are any better than the zoo of results provided in that paper. Other than that, I have only some more minor issues that I would like to see corrected or commented (see below). Therefore, I recommend publication after minor revisions.

Specific comments: Abstract, ln 27: Given all the uncertainties that the authors discuss themselves, I recommend not to end the abstract with this statement.

Experiments: Ln 134 ff: I suggest to discuss already here why you have chosen the piControl runs and not the historical simulations (I assume to have better statistic in the long runs). This needs to be justified as several studies have pointed to the forced component of the AMOC and AMV in the 20th century.

ln 140: which of the MPI models? Ln 141: there is no IPSL model in table 1

Ln 155 ff: EN4 should be introduced here and not in a half sentence when it is first used.

Labrador sea density: Ln 196 ff (Figure 1 b): it is interesting that some models show

very cold temperatures near the bottom. Is this originating from the overflows? Are these models specifically good in representing the overflows or include a parameterisations (CESM)? Do these differences matter in the later analyses?

Figure 1 a: I suggest to use a more intuitive and color-blind friendlier color map.

Figure 2b: it is not immediately clear that the orange and red curves belong together (one might think they are from MRI as in the previous figure), the thick green looks more like black in my print-out.

Figure 3: why is the 10 year time scale highlighted?

Figure 4d: what is the difference between this and figure 6 a?

Ln 340ff: if you want to explain cooling or warming, shouldn't one look at heat transport divergence/convergence?

Ln 351: could it also be related to a dynamical spin-up or down due to variations in the horizontal density structure (e.g. Born et al, 2013).

Figure 8: If the DePreSys AMOC is so far off from RAPID, why should it give a good observational estimate at 45N?

The symbols here and in fig 9 are very hard to decipher, e.g. I can't see GC2 in 8b

Figure 9: caption: it is hard to see any green line. Is there an "and" missing between "assimilation run" and "EN4"?

Response: The blue line was mistakenly tagged as green, and EN4 should have been included between brackets. Both errors have been corrected.

Ln 410ff: could the DePreSys run help to define which depth level of the DWBC is most consistent with observations?

---

## Author Comment (AC1) · 8 Feb 2021

**Response to Anonymous Referee 2**

**General Comment:**

Variations of the Atlantic Meridional Overturning Circulation (AMOC) and the associated meridional heat and fresh water transports are an important driver of climate variability in the North Atlantic region and beyond. They are also thought to provide a source of predictability on decadal time scales. Previous studies have described mechanisms behind this variability often focusing on the source regions of North At-

lantic Deep Waters. The authors describe the AMOC characteristics of about 20 CMIP5 models, most of them at resolutions in the ocean models of 0.5 to 1 degree, and two coupled models using an eddy-permitting ocean grid. They relate AMOC variability in the sub-polar and sub-tropical Atlantic to density changes in the Labrador Sea (LS). Such a connection has been invoked in several earlier studies, but the merit of this is that they provide an investigation in a multi-model context. They concentrate on the buoyancy-driven part of the overturning circulation and use long preindustrial control simulations to provide statistically sound results. A striking result is that there is a high degree of coherence in the models (at least in this specific class of models) regarding the relation between the LS density structure and the sub-polar North Atlantic circulation. Even the corresponding relation to the lower latitudes is still quite robust, even though model differences lead to different strengths of the correlations. The study itself and the individual analyses are well thought-through and the text describes adequately the results presented in the figures. Apart for sharing the critique of the first reviewer on missing discussions of some important previous papers (I would also include, for example, Delworth and Zeng (2016) where the mechanisms are nicely described), I feel that the authors overstate a bit the robustness of the "observational constraints". For example, given the results of Jackson et al. (2019), I don't think that the numbers of AMOC strength from DePreSys (used in figure 8) are any better than the zoo of results provided in that paper. Other than that, I have only some more minor issues that I would like to see corrected or commented (see below). Therefore, I recommend publication after minor revisions.

Response: We thank the reviewer for the time devoted to reviewing the article, and for all the good points raised, which we have addressed carefully to improve the quality of the manuscript. In the new version, we have extended the discussion of recent literature, as requested by the two reviewers, and expanded the discussion on the limitations related to the observational datasets used in this study, where we recommend further

work to assess the validity of the observation-based constraints (lines 585-591[1]).

The detailed answers to all the different comments, explaining how they have been addressed in the new manuscript, are included below. Also, a revised version of the article, highlighting the modifications with respect to its original version will be uploaded in the system after the editor's approval.

**Specific comments:**

Abstract, ln 27: Given all the uncertainties that the authors discuss themselves, I recommend not to end the abstract with this statement.

Response: We have removed the final sentence from the abstract.

Experiments: Ln 134 ff: I suggest to discuss already here why you have chosen the piControl runs and not the historical simulations (I assume to have better statistics in the long runs). This needs to be justified as several studies have pointed to the forced component of the AMOC and AMV in the 20th century.

Response: We now provide an explanation in the text (lines 160-163) of the reasons to use the piControl runs over the historical simulations, which includes the computation of more robust statistics thanks to the availability of longer experiments, and also an easier interpretation of correlations thanks to the absence of forced trends, which can lead to spurious significant linear relationships.

ln 140: which of the MPI models? Ln 141: there is no IPSL model in table 1

Response: This part has been rewritten. IPSL was the model with the coarsest effective resolution in Table S1 of Menary et al (2015), but it had not been included in this study because it lacked the necessary variables. We now mention three models in its place (GISS-E2-R/GISS-E2-R-CC/CanESM2), as the three have the coarsest resolution used in this study. Also, we have also noticed that the same table in Menary et al
* * *
[1]All line numbers refer to the version of the revised document with track changes

(2015) included GC2, which has a finer resolution than all the MPI models previously mentioned, and have changed it accordingly in the manuscript.

*Menary, M. B., Hodson, D. L. R., Robson, J. I., Sutton, R. T., Wood, R. A. and Hunt, J. A.: Exploring the impact of CMIP5 model biases on the simulation of North Atlantic decadal variability, GRL, 42(14), 5926–5934, doi:10.1002/2015GL064360, 2015.*

Ln 155 ff: EN4 should be introduced here and not in a half sentence when it is first used.

Response: This is a good suggestion. EN4 is now referenced here.

Labrador sea density: Ln 196 ff (Figure 1 b): it is interesting that some models show very cold temperatures near the bottom. Is this originating from the overflows? Are these models specifically good in representing the overflows or include a parameterisations (CESM)? Do these differences matter in the later analyses?

Response: The figure included in this response shows that the three models exhibiting very cold temperatures near the bottom (4000m and deeper) in Figure 1b are CanESM, GISS-E2-R-CC and GISS-E2-R. The four CESM model simulations (that as far as we know are the only model configurations with parameterised overflows; Danabasoglu et al., 2010; 2012) lie within the intermodel spread, which suggests that explicitly resolving the overflows (as opposed to parameterising them) does not necessarily affect temperature stratification in the Labrador Sea. It's also worth noting that the four CESM simulations show very high differences across them regarding the magnitude of the AMOC26 vs PC1-LSD cross-correlations. This can be inferred from Figure 8b, which points to other key contributions unrelated to the overflows to explain those large inter-model differences.

We cannot discard, however, that in some specific models, like GISS-E2-R-CC and GISS-E2-R, the overflows have a decisive impact in the deeper ocean levels, and through them in the LSD vs AMOC26 relationship, as both models are within the ones

supporting a stronger correlation between both variables (e.g. see Figure 8b). However, these models are not necessarily more realistic, as the very cold temperatures they present near the bottom are not supported by the EN4.2.1 observations. A detailed examination of the role of the overflows would take considerable time and effort, and we have preferred not to do it. Instead, we have included a discussion (lines 473-477) on the possible reasons behind the differences in density stratification across models, mentioning the potential role of the overflows.

*Danabasoglu, G., W. G. Large, and B. P. Briegleb, 2010: Climate impacts of parameterized Nordic sea overflows. J. Geophys. Res., 115, C11005, doi:10.1029/2010JC006243.*

*Danabasoglu, G., S. Bates, , B. P. Briegleb, , S. R. Jayne, , M. Jochum, , W. G. Large, , S. Peacock, , and S. G.Yeager, 2012: The CCSM4 ocean component. J. Climate, 25, 1361–1389.*

Figure 1 a: I suggest to use a more intuitive and color-blind friendlier color map.

Response: The color palette has been changed to viridis (https://www.thinkingondata.com/something-about-viridis-library/) that is color-blind friendly. Also, in the new version of Figure 1a darker colors represent higher differences in density, to make it more intuitive.

Figure 2b: it is not immediately clear that the orange and red curves belong together (one might think they are from MRI as in the previous figure), the thick green looks more like black in my print-out.

Response: The colors in the timeseries of the decadal running trends are now only slightly darker than for their respective raw time series, to make it more clear that they represent the same simulation.

Figure 3: why is the 10 year time scale highlighted?

Response: It is highlighted to separate the interannual from the decadal/multi-decadal

timescales. This is now indicated in the figure caption.

Figure 4d: what is the difference between this and figure 6 a?

Response: They are the same. This panel had been included in both figures by mistake. Figure 4d has been removed.

Ln 340ff: if you want to explain cooling or warming, shouldn't one look at heat transport divergence/convergence?

Response: Our goal with Figure 7 is not so much to explain the ESPNA T700 changes in terms of heat transport convergence/divergence, as a sort of heat budget analysis, but to link it to changes in the heat transport that can be directly attributed to AMOC variability.

Ln 351: could it also be related to a dynamical spin-up or down due to variations in the horizontal density structure (e.g. Born et al, 2013).

Response: This dynamical spin-up could explain a delay in the response of the SPG, but this is not supported by Figure 4c, showing that PC1-LSD and the SPG are in phase. We don't see how the changes in the horizontal density structure (which would mostly affect the SPG strength) could explain a delay in the OHT-gyre component. For us the most likely explanation is the delayed advection of temperature anomalies, already mentioned in the manuscript.

Figure 8: If the DePreSys AMOC is so far off from RAPID, why should it give a good observational estimate at 45N?

Response: This is a very good point. We now acknowledge this difference between DePreSys and RAPID in the text (lines 449-452), and recommend caution when using DePreSys as an observational reference for the AMOC45.

The symbols here and in fig 9 are very hard to decipher, e.g. I can't see GC2 in 8b

Response: We have changed the size of the symbols, and the color palette to improve

the visibility.

Figure 9: caption: it is hard to see any green line. Is there an "and" missing between "assimilation run" and "EN4"?

Response: The blue line was mistakenly tagged as green, and EN4 should have been included between brackets. Both errors have been corrected.

Ln 410ff: could the DePreSys run help to define which depth level of the DWBC is most consistent with observations?

Response: It can. Indeed, the depths at which the largest correlations between PC1-LSD and the WBDs are attained in DePreSys3 are provided as vertical dashed lines in Figure 11. We were not previously discussing these DePreSys values in the article, but we do so in the new version of the manuscript (lines 521-523)
* * *
[Figure]

[Figure]

**Fig. 1.** Climatological mean of the spatially averaged Labrador Sea (blue box in panel a) temperature as a function of depth in the simulation ensemble, the DePreSys3 assimilation run and EN4.

---

## Author Comment (AC2) · 8 Feb 2021

**Response to Anonymous Referee 1**

**1. General comments**

Ortega et al. analyze the internal multidecadal variability of the CMIP5 piControl ensemble (~1° horizontal resolution) and two additional model simulations with a higher resolution (1/3° and 1/4°). Cross-correlations show significant links between subsurface density variability in the western subpolar North Atlantic (represented by the principal component of density; PC1-LSD) and the AMOC at different latitudes when the

Ekman component is removed. The correlation is not coherent over different latitudes and the driving factors of the relationship are derived.

The manuscript provides a new and certainly helpful index for density anomalies in high latitudes and the multi-model perspective offers robust insights into the internal variability of coupled climate models. The text is well written and good to follow. I am not very surprised by most of the findings as in the context of Atlantic Multidecadal Variability (AMV) analyses of various models showed similar results (see point 3 in the next section). The discussion of these is almost missing. Those and other issues are described in detail below. Once these are addressed, I would like to see the manuscript published as it improves our ability to "predict earth system change" (from the scope text of ESD).

Response: We thank the reviewer for the time devoted to reviewing the article, and for all the good points raised, which we have addressed carefully to improve the quality of the manuscript. We have now extended the discussion of recent literature, including the recent findings from multi-model AMV comparisons.

The detailed answers to all the different comments, with an explanation of how they have been addressed in the new manuscript, are included below. Also, a new version of the article, highlighting the modifications with respect to its original version will be uploaded in the system together with the revised version.

**2. Specific comments**

The main rationale of this paper is the interpretation of cross-correlations of processes that exhibit a red frequency spectrum. Large significant correlations are found between (subsurface) density anomalies/variability in the western subpolar North Atlantic (spatial average over the Irminger Sea, Labrador Sea and the boundary current along Greenland; blue box in Fig. 1a) and the subpolar gyre circulation strength in the same area (SPGSI) and the AMOC at 45°N (AMOC45). This link decreases with distance as shown for the subtropical large-scale circulation, represented by the AMOC at 26°

ESDD
N (AMOC26). I am not very surprised by these results: 1) The active tracers temperature and salinity are transported to the subpolar gyre (SPG) by the strong Gulf Stream and the North Atlantic Current. Both tracers change density and hence the circulation. The stronger the circulation, the stronger the tracer transport/input. In this very general sense, Fig. S1 e and f are nearly identical: here, it does not matter if one correlates the SPG strength with the deep Labrador Sea density itself (dLSD) or with its first principal component (PC1-LSD). Then, by definition, the rather red frequency spectrum yields smaller correlations with increasing lag and/or space. Based on this I would expect lower correlations with the remote AMOC26 compared to AMOC45. Density anomalies propagate with the Western Boundary Current (WBC) – or the western boundary densities (WBD) as used here – but model differences in re-circulations, current strength and location or diffusion yield a larger model spread on the long way to the subtropics.

Response: We completely agree with the interpretation of the cross-correlations in Supplementary Figure 1 (and also in Figure 4), including the reasons for the differences seen between the PC1-LSD vs AMOC26 and the PC1-LSD vs AMOC45 plots. Both the stronger correlations and the narrower multi-model spread in the PC1-LSD vs AMOC45 cross-correlations (as compared to AMOC26) are foreseeable results. We have included a new sentence (lines 336-338-1) to explain why these results are expected. In our view, even if the larger inter-model uncertainties between PC1-LSD and AMOC26 are not surprising, it is important to document and understand them. Indeed, the most original aspect of the article is the subsequent analysis in which the PC1-LSD vs AMOC26 inter-model differences are investigated, and linked back to differences in specific model properties (e.g. mean strength of the ocean circulations, representation of boundary densities).

Since you remove the Ekman component from the AMOC indices, short-term variability is removed. I speculate that if the Ekman component would be included, the spectra
<sup>1All line numbers refer to the version of the revised document with track changes

would be closer to white noise (e.g. Figure 2 d of Ba et al. 2013) and the correlations would decrease more rapidly due to the more short-term wind-induced fluctuations (Wunsch and Heimbach 2012). If this is the case, does the main finding, the multi-decadal variability in the Labrador Sea density as a precursor of the AMOC, applies also in the real world (or at least in the models) with Ekman contribution? If not, what do we learn instead? The role of the Ekman component should already be mentioned in the abstract.

Response: The reviewer is right, by removing the Ekman component the power spectra of AMOC indices become redder, as shown in the new Supplementary Figure 1. The Ekman component inclusion introduces moderate changes on the PC1-LSD vs AMOC cross-correlations (New Supplementary Figure 2), although the main finding still holds for most models: that is, that the PC1-LSD index is generally a good precursor of the AMOC changes (with or without Ekman). This is probably because despite the high-frequency fluctuations of the Ekman component, the full AMOC (i.e. with Ekman) variability is still dominated by the long-term signals, as indicated by the fact that its spectrum corresponds to the one of a red-noise process. The main effect of Ekman, in particular for the AMOC26, is a modification in the lag at which its correlation with PC1-LSD is maximum, which in some models becomes negative (i.e. the AMOC26 leads the PC1-LSD). This apparently counterintuitive lag-relationship is related to the fact that NAO-driven surface zonal winds exert two different influences on the AMOC: they contribute to instantaneous changes via its Ekman component, and also to lagged changes through its cumulative influence on the LSD, which is latter advected southward inducing a thermal wind response. In some models (e.g. HIGEM), the instantaneous wind contribution appears to be dominating the PC1-LSD correlations with the AMOC, and because the PC1-LSD responds with some lag to the NAO wind-driven forcing (see Figure 3a in Ortega et al., 2017), the maximum correlations occur when the AMOC index leads. This effect hampers the physical interpretation of the results, and is one of the reasons that led us to remove the Ekman component from the AMOC indices. The other reason is that by removing the Ekman component
(which is mostly associated with atmospheric noise and therefore unpredictable) we could focus on the slow wind-forced and thermohaline-driven AMOC variability, that is the one expected to be more closely linked to the LSDs. We have expanded different parts of the manuscript to discuss the sensitivity of the results to the inclusion/exclusion of the Ekman signal (lines 343-351). The sensitivity to the Ekman component is also mentioned now in the abstract (lines 24-25).

Similar findings were described in the context of the Atlantic Multidecadal Variability (AMV). As you refer to Knight et al. 2005 in the introduction, I think the findings should be discussed with those studies that relate local (subsurface) hydrography and large-scale ocean circulation variability (e.g. Polyakov et al. 2010, Ba et al. 2013, or preferably with section 3 of the review of Zhang et al. 2019). This is of particular interest since the CMIP5 ensemble tends to underestimate internal variability (Cheung et al. 2017).

Response: We have expanded the introduction to discuss previous studies investigating the potential mechanisms linking the AMOC with the AMV and the SPNA variability (lines 54-58 and 115-117). And we have also expanded the final section (lines 591-598) to now discuss several aspects of the AMV that are differently represented across-models (with the respective studies), and their potential link with our findings.

In addition, I miss several points in the discussion/conclusions:

The box chosen for PC1-LSD is rather large. From my experience, the signal will be dominated by the WBC and is possibly quite different from the Irminger or Labrador Sea interiors. Due to this choice, it is difficult to address some of the uncertainties that you summarize in the introduction (L93f). Wouldn't this study be a good opportunity to test different locations by calculating e.g. PC1-IS for the Irminger Sea?

Response: We decided to use a large box to represent the western SPG to make sure that the main area of North Atlantic deep convection was included. Deep convection happens predominantly in some models in the Labrador Sea interior, and in others in
the Irminger Sea, and has a dominant role on the local densities. This large-box also allows us to follow-up on previous studies based on individual experiments (Ortega et al 2017; Robson et al 2016) that linked western SPG density variability with the changes in the wider North Atlantic. Understanding the contributions from the different sites (Labrador vs Irminger) and the differences between them would require further analyses and open new questions, that we prefer to leave for future work. Indeed, we envisage to investigate inter-site differences across models, evaluating them against observations from the OSNAP array, in a follow-up study.

Ortega, P., Robson, J., Sutton, R. T. and Andrews, M. B.: Mechanisms of decadal variability in the Labrador Sea and the wider North Atlantic in a high-resolution climate model, Clim. Dyn., 49(7–8), doi:10.1007/s00382-016-3467-y, 2017.

Robson, J., Ortega, P. and Sutton, R.: A reversal of climatic trends in the North Atlantic since 2005, Nat. Geosci., 9(7), doi:10.1038/ngeo2727, 2016.

Likewise, it would be great if you could redo the calculations with the historical CMIP5 ensemble in addition to the pure internal variability of the picontrol runs. I am aware that this would be beyond the scope of the paper as it is at the moment, however, if you and your colleagues discuss the work further, I would really like to see these runs included.

Response: This is a very good suggestion, that would open new interesting questions. But it would also bring new limitations, the most important being time. Repeating the analysis with the historical experiments would imply an important effort to download, postprocess and analyse large quantities of data, and would also require to re-structure the paper and the figure selection to accommodate the new simulations and results. Besides, historical simulations come with forced trends, which need to be dealt with carefully, in particular for investigating linear relationships (as we do in the article) to avoid drawing spurious conclusions, like nicely illustrated in Tandon and Kushner (2015). Also, since different models have different climate sensitivities, the external
forcings would introduce an additional source of uncertainty in the relationships. All these aspects deserve to be conveniently investigated and discussed, and we prefer to leave it for future work.

Tandon, N. F., and Kushner, P. J.: Does External Forcing Interfere with the AMOC's Influence on North Atlantic Sea Surface Temperature?, Journal of Climate, 28, 6309-6323. 2015.

I miss a final statement if and why PC1-LSD is better suited than e.g. dLSD or any other measure for deep water formation activity? What kind of index should be used by the community? Can you give a recommendation?

Response: We have included a final statement with a recommendation on lines 362-366.

**3. Technical corrections:**

Please state the version of EN4 you used somewhere in the main text (not only in the url in "Data availability").

Response: The data version of EN4 (2.1) was indicated in the text of Data availability section, together with the url. For clarity, we now refer to it as EN4.2.1 every time it appears in the main text and in the figures.

Regarding all scatter plots: it is a bit difficult for me to distinguish the yellow/light green symbols representing the different models (I need to zoom in with the pdf viewer, don't know how it looks printed). If you want to stick to the colors, you could use capital letters instead of the symbols?

Response: We have increased the size of the symbols in the scatter plot, and also changed the color palette to one without light color tones.

The references are very annoying to check due to missing line breaks.

Response: We have included line breaks as suggested, and also indented the first line
in each reference so that all references are more clearly separated.

L134f L139f: Can you give the number of model years of the picontrol and present day simulations?

Response: The model years are included in Table 1. This is now explicitly mentioned the first time the table is cited (line 166).

L141: Rephrase "IPSL" so that one can find the model in Table 1.

Response: Thanks for spotting this problem. We had wrongly referenced IPSL, the model with the coarsest effective horizontal resolution in Menary et al. (2015) but that couldn't be included in our multi-model analysis because it didn't provide the necessary data. We have also noticed now that GC2 was also included in Menary et al. (2015), and has the finest resolution of all models (including the MPI versions, which were mentioned in the earlier version). The manuscript has been corrected accordingly.

Menary, M. B., Hodson, D. L. R., Robson, J. I., Sutton, R. T., Wood, R. A. and Hunt, J. A.: Exploring the impact of CMIP5 model biases on the simulation of North Atlantic decadal variability, GRL, 42(14), 5926–5934, doi:10.1002/2015GL064360, 2015.

L149f: Can you briefly describe the main differences between picontrol and present day experiments) one sentence enough)?

Response: The main difference is that the picontrol experiments have fixed radiative forcing conditions (e.g. GHG concentrations, aerosol and ozone concentrations,...) representative of year 1850, while the present day experiment has also fixed radiative forcing conditions but from year 1990. This information is now provided in lines 159-160, and line 175.

L168: EOS-80 is deprecated. TEOS-10 should be used in the future.

Response: We thank the reviewer for the advice and will use equation TEOS-10 in future analyses.

**ESDD**
L179f: I would like to see the 10-year running mean time series in addition to the running trends in Fig. 2b.

Response: We have added the 10-year running means to the figure as thin dashed lines.

L211f: Most models are too dense by > 0.1 kg m-3 at the surface and > 0.2 kg m-3 at the subsurface as shown in many previous papers. In my view, "good representation" is too optimistic.

Response: We have substituted "good representation" by "comparatively better representation".

L215f: Please give a reference on how you calculated the EOF (there is also no ref in Ortega et al. 2017).

Response: A reference to Storch and Zwiers (1999) has been included, an article in which the methodology to compute EOFs and the use of EOFs in climate research are widely described.

Storch, H., Zwiers, F. (1999). Statistical Analysis in Climate Research. Cambridge: Cambridge University Press. doi:10.1017/CBO9780511612336

L218f: It's not clear what it means when the "vertical profile in Figure 1 is weakened or strengthened". Can you clarify this related to stratification? Does this apply to the temp/salt profiles as well?

Response: In the sentence we referred only to density, that is the variable described by the EOF. We meant to say that stratification, as described by the mean vertical profile of density, is weakened or strengthened for positive/negative values of the principal component. We have rephrased the sentence for clarity.

L223: In what way is the multidecadal variability seen in Fig. 2b "important"?

Response: The term "important" has been substituted by "clear".
L226: This sentence confuses since the deep LSD was not mentioned before. Why not something like "In addition to the PC1-LSD index we define the deep LSD (dLSD) as . . . "?

**Response: We have changed the sentence following the reviewer's suggestion.**

L237f: Add to the caption that  $\sigma$ 2 density is used at depth. Is the sigma symbol missing in "(2)"? Can you add axes to Fig. 1a?

Response: The caption has been corrected, and the axes to Fig. 1a have been added.

L245f: Is it possible to add calendar years e.g. as an upper x-axis to Fig. 2b since one of the runs represent present day? Is there a reason why you don't show a PC1-LSD time series of one (or more) of the CMIP5 models in Fig. 2b?

Response: The HIGEM simulation is a present-day control simulation, which means that it uses fixed radiative forcing conditions that are representative of the present climate. It does not follow the historical evolution of the forcings, and therefore does not represent calendar years. We didn't include any CMIP5 simulation in Figure 2b for the sake of visibility (i.e. to avoid the figure becoming too busy). For comparing the timescales across models we already have Figure 2c, that includes all simulations.

L266: Please state the depth d that you use as the Ekman depth since Baehr et al. 2004 give a range (50-100 m).

Response: We used a fixed Ekman depth of 50 m, which is now mentioned in the text.

L271f: By chance, could you name the main differences of the Fourier spectra if the Ekman is included (one sentence enough)?

Response: These differences are now mentioned in lines 323-325.

L278: AMOC45 and AMOC26 and not AMOC45N and AMOC26N.

Response: Corrected.
L280f: Please also write at least one sentence to Fig. 4d (or remove Fig. 4d since Fig. 4d = Fig. 6a).

**Response: We have removed the Figure 4d panel.**

L292: You write that the cross-correlation lags in Fig. 5 are up to 10 years. In the caption of Fig. 5 you write that the lags can be any (for me, this indicates that lags > 10 years are possible). Can you please clarify that in the text and/or caption?

Response: We have only computed the correlations up to the 10 year lags. We clarify this now in the caption of Figure 5.

L296f: Can you speculate why the model spread increases when dLSD is used compared to PC1-LSD? Also, in my view, the model spread increases only for the AMOC indices. The cross-correlations with the SPGSI and ESPNA-T700 indices are more or less independent of the choice of LS density index (Fig. S1).

Response: dLSD concentrates on variability at deeper levels, and is therefore more sensitive to influences from the Arctic overflows, which are also linked to the AMOC and can be very differently represented across models, introducing a new source of uncertainty. This is now mentioned in the text (lines 360-362). We also specify now that the increase in spread in dLSD with respect to PC1-LSD only happens for the AMOC plots (line 360).

L312: ". . . 2005 to 2015 (Robson et al. 2016; Ortega et al. 2017)"  $\rightarrow$  ". . . 2005 to 2014 (Robson et al. 2016)"

Response: Corrected.

L314f: Put the ESPNA-T700 definition to section 3.2, where you describe Fig. 4 (Fig. 4d = Fig. 6a).

Response: Since we have removed the panel d in Figure 4, it makes sense to keep the ESPNA-T700 definition here.

ESDD
L314: "temperature"  $\rightarrow$  "potential temperature"

Response: Corrected.

L326f: The ocean plays a role in what? This sentence is a little strange to me (of course, the ocean plays a role?).

Response: We have rephrased the sentence (lines 398-399).

L336: Correct "of the first on the".

Response: The sentence has been rewritten.

L377f: Do you take the maximum cross-correlations of PC1-LSD and AMOC26 or the maximum of the absolute values of the cross-correlations? It is difficult to judge from Fig. 4b but maybe you can clarify that all the maximum cross-correlations occur when PC1-LSD leads AMOC26 (if this is the case).

Response: We take the maximum of the correlations (regardless of the sign) at any lag in which PC1-LSD leads AMOC26. These maximum correlations are always positive, as can be seen in Figure 4b. We changed the figure caption to explain it better.

L378: AMOC45 and AMOC26 and not AMOC45N and AMOC26N.

Response: Corrected.

L389f: The sentences "Models with . . ." and "Models that have ..." have the same content. Remove one of them.

Response: Thanks for noticing the redundant content. We have removed the first sentence.

L393(1): In Fig. 9, depth averages from 0-100 vs 500-1000 m were used. In the header of Fig. S2b as well as in the caption of Fig. S2 is written that depth averages from 0-100 vs 500-1000 m were used. I am confused since these are the same depth ranges but for you derive different results in Fig. 9c compared to Fig. S2b.
Response: Figure 9c and Figure S2b are the same. We have repeated that panel in Figure S2b to ease the comparison with the results from other depth ranges. We now specify in the caption of Figure S2b that the second panel is the same as in Figure 9c, to avoid confusions.

L393(2): I am not totally convinced by the stratification definition used in Fig. 9 and Fig. S2. From Fig. 9 a and b you infer that "no link has been found between the PC1-LSD vs AMOC26 relationship and temperature and salinity stratification in the Labrador Sea" (L390). Assuming that you indeed find significant and non-significant correlations for the same variable depending on the depth range, I would rephrase the sentence from "no T/S link" to something like "weaker T/S link than density … depending on depth range".

Response: We have checked the corresponding correlations for the three Labrador Sea stratification indices used in Supplementary Figure 4 (old Supplementary Figure 2), but defined in terms of salinity and temperature. All these correlations are weaker than those computed for the corresponding density stratification index, and only in one case the correlation is significant at the 95% confidence level (i.e. for the index comparing salinity between 0-200m with salinity between 1000-2000m). For the temperature stratification indices, and for the salinity index defined at other depth ranges the correlations are consistently weak and insignificant. Because of that, we have changed the sentence to "No robust link between the PC1-LSD vs AMOC relationship and both temperature and salinity stratification in the Labrador Sea has been found".

L403: green  $\rightarrow$  blue; EN4  $\rightarrow$  (EN4)

Response: Corrected.

L405: Add to the caption that  $\sigma$ 2 density is used at depth.

Response: Added to the caption.

L407: LSD  $\rightarrow$  PC1-LSD; Add " (WBD)" at the end of the sentence.
Response: Both changes have been included.

L412: "It represents the correlations"  $\rightarrow$  "It represents the in-phase correlations" (or lag-0 or . . .)

Response: Changed accordingly.

L413f: Please add the section lines to Fig. 1a and state that  $\sigma$ 2 density is used.

Response: The latitudinal cross-section lines have been added to Fig. 1a. We also indicate now in the text that density is computed as sigma 2.

L421: "Figure 10f"  $\rightarrow$  "Figure 10g"

Response: Corrected.

L426: "HiGEM 57"  $\rightarrow$  "HiGEM 57N" in header of Fig. 10e.

Response: Corrected.

L434: I find the formulation "deeper WBDs" a little misleading since its the correlation between PC1-LSD and the zonal density field which is deeper in those models, not the WBD itself (in the sense of a boundary current).

Response: The reviewer is right. We have rephrased the sentence to "models exhibiting maximum correlations with the WBDs at deeper levels generally show stronger links between PC1-LSD and the subtropical AMOC".

L435: The sentence "We also checked . . ." can only be understood together with the caption of Fig. 11. Please clarify.

Response: The sentence has been rewritten.

L440 L444: "AMOC26N"  $\rightarrow$  "AMOC26"

Response: Corrected.

L444: "57N"  $\rightarrow$  "57° N"
Response: Corrected.

L446: "run ."  $\rightarrow$  "run." and "WDB"  $\rightarrow$  "WBD"

Response: Both corrected.

L462f: The second and third findings can be merged.

Response: We prefer to keep them separate, so that all findings have a similar size.

L481: "wstern"

Response: Corrected.

L505: Given the high latitudes I would not say that two 1/3° and 1/4° horizontal resolution ocean models with a similar climatology as the 1° CMIP5 ensemble imply that the representation of the mesoscale might not be as important as the climatology itself. The first baroclinic Rossby radius is  $\hat{a}$ Lij10 km or lower and as you give with the 1/12° reference, resolution does matter.

Response: We have removed the last part of the sentence, in which we were making that claim.

L510: Is it really "including in" or just "including"?

Response: We have substituted "including in" with "affecting", which makes a clearer sentence.

Caption of Fig. S1: ". ."  $\rightarrow$  ". "

Response: Corrected.

---

## Author Response (AR2)

**Response to Anonymous Referee #1**

**Specific comments**

L339ff: The influence of the Ekman component on the subtropical AMOC is much stronger in the GC2 model compared to all other models (green lines in Fig. S2c and d). Can you speculate about the reasons?

Response: We believe that it is related to the same problem discussed later on in the paragraph, that is, that NAO associated winds induce an instantaneous increase of the subtropical AMOC and at the same time exert a delayed influence on the PC1-LSD, which also affects the subtropical AMOC with a delay. The final correlations between PC1-LSD and the full AMOC (i.e. with Ekman) depend on how quickly the PC1-LSD propagates to the subtropics and impacts the AMOC, and how this interferes with the Ekman driven signal. We can see in Figure S2c that the delay of the maximum correlation between PC1-LSD and AMOC26 is particularly long in GC2 and rather weak when compared with the other models with a similarly long lag. These two factors might determine why the interference between the two signals has such a strong impact in GC2. The specific case of GC2 is now discussed in the paragraph.

**Technical corrections**

Please check all $\sigma$2 and correct them to $\sigma_2$.

Response: Done

L256: Figure 1d, not 1c. In general, the signs of the result of an EOF analysis are arbitrary. How do you know that PC > 0 corresponds to a decreased density stratification and not the other way around? If you infer that from the positive relationship between PC1-LSD and deep LSD from Fig. 2d, then please mention it in the text.

Response: The reference to Figure 1d has been corrected. Regarding the sign of the EOFs, we have taken advantage of the fact that it is arbitrary (as well pointed out by the reviewer) to describe all EOFs as an intensification of the density stratification. This implied changing their sign (together with the sign of their associated PC) in the individual cases in which the EOF originally computed represented a decrease in stratification. This is now specified in the caption to Figure 2.